# ON THE HARDNESS OF FAITHFUL CHAIN-OF-THOUGHT REASONING IN LARGE LANGUAGE MODELS

## ABSTRACT

As Large Language Models (LLMs) are being increasingly employed in critical domains such as healthcare, it is essential to make these models trustworthy. In this pursuit, Chain-of-Thought (CoT) prompting has emerged as a potential source of transparency in LLMs. While CoT reasoning is appealing to humans, prior studies have shown that these reasoning chains are not faithful i.e.; they do not accurately reflect the underlying LLM's behavior. Ensuring the faithfulness of LLM-generated CoT reasoning is crucial for decision-makers, who rely on them to determine if, when, and to what extent, trust the recommendations made by these models. While several works proposed strategies to enhance accuracy and truthfulness in LLMs, there has been a lack of exploration on the effectiveness of these common strategies to enhance the faithfulness of chain-of-thought (CoT) reasoning. Specifically, we explore the promise of in-context learning, fine-tuning, and activation editing to improve the faithfulness of the CoT reasoning. Our empirical analyses on benchmark tasks indicate that these strategies offer limited success in improving the faithfulness of the CoT reasoning, with only slight performance enhancements in controlled scenarios. Activation editing demonstrated minimal success, while fine-tuning and in-context learning achieved marginal improvements that failed to generalize across reasoning and truthful question-answering benchmarks. We subsequently analyse what makes faithful CoT reasoning challenging, and present findings to lay the groundwork for future research in trustworthy reasoning from LLMs. In summary, our work underscores the inherent difficulty in eliciting faithful CoT reasoning from LLMs, suggesting that the current array of approaches may not be sufficient to address this challenge.

## 1 INTRODUCTION

Large Language Models (LLMs) are increasingly being employed in diverse real-world applications ranging from content generation and education to commerce and healthcare (Kaddour et al., 2023). One of the primary reasons behind the widespread adoption of these models is their enhanced reasoning capabilities, which enable them to generate responses that appeal to human end users (Brown et al., 2020b; Wei et al., 2022b). Furthermore, these models are also capable of explaining the rationale behind the responses they generate, in a manner that is appealing to humans. Despite the aforementioned advantages, LLMs also suffer from some critical drawbacks. For instance, while LLMs are adept at producing explanations that cater to human preferences, recent research (Lanham et al., 2023; Turpin et al., 2023) has demonstrated that the explanations generated by these models – *e.g.,* Chain-of-Thought (CoT) reasoning – do not *faithfully* capture their underlying behavior. The faithfulness of the generated explanations turns out to be an important desideratum in high-stakes applications such as medical diagnostics and legal counseling. Ensuring the faithfulness of LLM-generated CoT reasoning is crucial for decision-makers, such as doctors, who rely on them to determine if, when, and how much to trust the recommendations made by these LLMs.

Despite the criticality of the faithfulness of LLM-generated reasoning, there is very little research on measuring and enhancing this aspect of LLMs. Recently, Lanham et al. (2023) introduced a slew of metrics for measuring the faithfulness of the CoT reasoning generated by LLMs. For instance, they propose an *early answering* faithfulness metric, which considers a generated CoT to be faithful if truncating that CoT causes the model to change its final response. While measuring the faithfulness of an LLM-generated CoT is one critical aspect, another piece of this puzzle is figuring out ways

to improve the faithfulness of the CoT reasoning generated by LLMs. While prior works have developed approaches to make CoT more aligned with human understanding or knowledge (Lyu et al., 2023), there are no solutions that focus on improving the faithfulness of LLM-generated CoTs in such a way that they accurately capture the behavior of the underlying model (please refer to Appendix for a more detailed discussion on related work). Furthermore, it remains unclear how difficult it is to improve the faithfulness of LLM-generated CoT reasoning.

**Present work.** In this work, we address the challenge of generating faithful CoT reasoning by exploring the promise of three broad approaches—activation editing, fine-tuning, and in-context learning—to enhance the faithfulness of the CoT reasoning generated by LLMs. Activation editing (Li et al., 2024) involves probing the internal structures of LLMs and strategically updating them to improve certain properties while fine-tuning leverages curated datasets to learn the task. In-context learning, on the other hand, involves providing a handful of samples to the model at inference time to tweak its behavior. These three approaches have found success in modifying LLM outputs to improve properties such as accuracy (Wei et al., 2022a), truthfulness (Li et al., 2024), tasks, reduction of biases and hallucinations (Tonmoy et al., 2024; Liu et al., 2024b), they have not been explored in the context of improving the faithfulness of LLM-generated CoT reasoning. This exploration is crucial and lays the groundwork for future research on improving the faithfulness of CoT reasoning in LLMs.

Here, we introduce novel activation editing, fine-tuning, and in-context learning strategies to improve the faithfulness of LLM-generated CoT reasoning. Specifically, we introduce an activation editing strategy that involves probing LLMs first to identify a vector/direction corresponding to faithfulness and then editing specific attention heads by translating along the identified faithfulness vector. Our fine-tuning and in-context learning strategies involve leveraging the metrics outlined in Lanham et al. (2023) to identify specific instances and their corresponding faithful CoT reasoning, and providing these as inputs to the LLM during the fine-tuning or in-context learning phases, respectively.

Despite the promise of these techniques, our findings reveal that none of them significantly enhance the faithfulness of the CoT reasoning generated by LLMs. While activation editing approach demonstrates limited success in amplifying faithful behavior of CoT reasoning, the fine-tuning and ICL approaches slightly improved CoT faithfulness in controlled scenarios but did not generalize well across diverse datasets. Our results underscore the inherent difficulty in eliciting faithful reasoning from LLMs, suggesting that the current array of techniques available to us is insufficient for addressing this complex challenge. Our research emphasizes the need for fundamentally new methodologies that can delve into the inner workings of LLMs to enhance the faithfulness of LLM-generated CoT reasoning, ensuring that LLMs are not only generating correct responses but also doing so in a manner that faithfully reflects their internal reasoning processes.

## 2 PRELIMINARIES

Next, we define the notion of faithfulness used to evaluate CoT reasoning from LLMs and then discuss mathematical notations that describe strategies for eliciting faithful reasoning from LLMs.

**Chain-of-Thought (CoT).** CoT reasoning in LLMs provides a structured response where the model explicitly generates the step-by-step thought process leading to its final response. This technique is particularly useful in complex reasoning tasks, such as solving math problems or logical question-answering scenarios, and high-stakes decision-making, where transparency in decision-making is crucial. By eliciting intermediate steps, CoT significantly improves the accuracy of LLMs on reasoning tasks and simultaneously leads to greater user trust and understanding. A relevant stakeholder can now see how the LLM processes the input information and relies on it to generate the final output response. See Fig. 1 for examples of CoT reasoning. This CoT reasoning can potentially make the LLM's reasoning process more transparent and easier to trust. Further, this also mimics human problem-solving approaches, allowing for easier debugging and refinement of model reasoning.

**Notations.** Formally, let $\mathcal{F} : Q \rightarrow A$ denote a large language model that maps a sequence of $n$ input tokens $Q = (q_1, q_2, \ldots, q_n)$ to sequence of $m$ answer tokens $A = (a_1, a_2, \ldots, a_n)$, where $q_i$ and $a_i$ are text tokens belonging to the model vocabulary $\mathcal{V}$. For CoT reasoning, we append the input tokens $Q$ with a prompt that follows the template: "*Read the question, give your answer by*

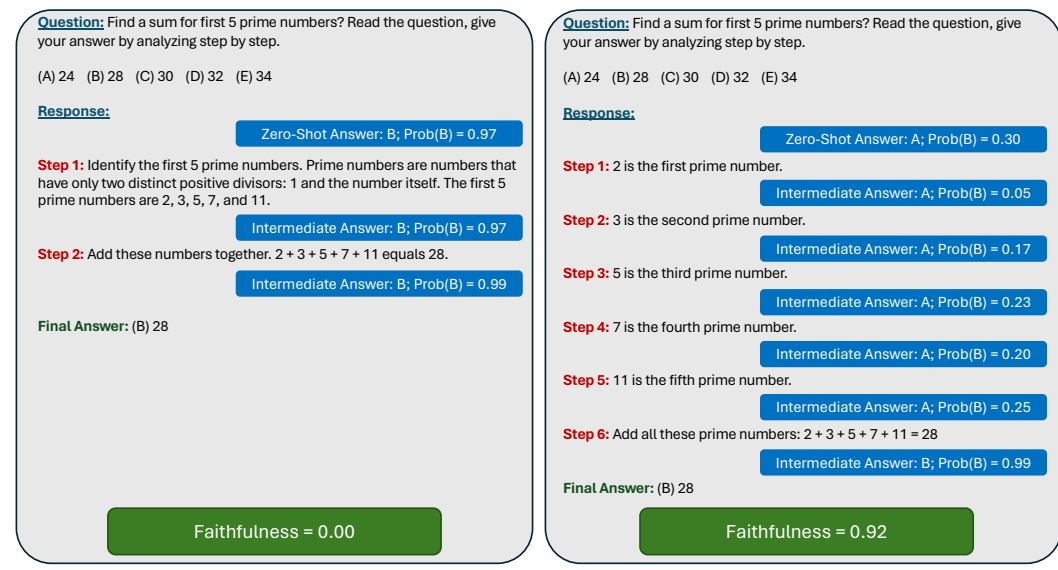

Figure 1: Examples for Unfaithful (left) and Faithful (right) explanations generated by state-of-the-art GPT-4 (left) and LLAMA-3-8B-INSTRUCT (right) LLMs. The faithfulness score is calculated using the early answering metric proposed in Lanham et al. (2023). A faithful CoT reasoning is one where prediction probability of the correct answer gradually improves with an increase in CoT steps.

*analyzing step by step, ...*". For the activation editing of LLMs, we train different linear classifiers $f : x \rightarrow y$, where $x \in \mathbb{R}^{d_{\text{head}}^l}$ are the intermediate layer activations of model $\mathcal{F}$ for a given input sequence $X$, $d_{\text{head}}^l$ is the dimension of the model activations at layer $l$ and attention head, and $y$ is the respective label associated with the input. We define sampling functions $S(\tau, p, \texttt{mode})$ and $S(\tau, \texttt{nshot}, \texttt{mode})$ that we use to sample different fine-tuning and in-context examples in our strategies in Sec. 3, where $\tau$ determines the temperature parameter of the LLM used to control the randomness in the generated answers by using the probability distribution of each generated token, $p$ denotes the percentage of training examples we use in fine-tuning, $\texttt{nshot}$ denotes the number of training examples we use in the ICL prompting, and $\texttt{mode}$ denotes the sampling technique, *i.e.,* whether we want to randomly sample examples from the train split or select the examples with most faithful explanation.

**Measuring Faithfulness.** While faithfulness is formally defined as how well an explanation accurately reflects the reasoning process of the underlying LLM, operationalizing this definition in the context of LLMs is non-trivial, partly due to the billion parameter scale, black-box nature of LLMs, and partly due to the internal reasoning (typically a combination of multiple complex nonlinear functions) being in a different representation space from textual CoT reasoning (Agarwal et al., 2024). We utilize faithfulness metrics proposed in Lanham et al. (2023) that quantify the faithfulness of CoT reasoning from black-box LLMs. Specifically, we employ the *Early Answering* strategy, which evaluates the faithfulness of a CoT by sequentially adding each CoT step to the question and querying the LLM for its answer, conditioned on the truncated set of CoT steps. If the answer from the LLM converges towards the final answer as it encounters more CoT steps, it indicates that the CoT explanation is guiding the answer and is more likely to be faithful. While there are other possible faithfulness measures, each has its own limitations detailed in Table 1 of the Appendix.

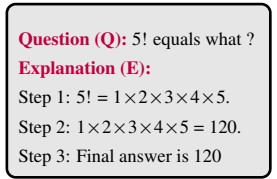

(a) Example of CoT reasoning

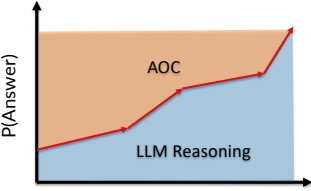

(b) Measuring faithfulness

To evaluate the faithfulness of a CoT $E$ shown in Fig. 2a, the early answering strategy involves providing different truncated versions of $E$ and analyzing how the LLM responds to it. For example, if we provide just the first step of $E$, *i.e.,* Prompt = "*5! equals what ? 1: 5! = 1×2×3×4×5.*" and the LLM does not return 120, but it returns 120 when provided with all the steps in $E$, *i.e.,* Prompt

= *"5! equals what ? Step 1: 5! = 1×2×3×4×5. Step 2: 1×2×3×4×5 = 120. Step 3: So the final answer is 120"*, then we can conclude that $E$ is likely to be faithful. Finally, faithfulness is quantified by the area over the curve (AOC) of explanation fraction vs. the percentage of answers consistent with a full explanation. Note that Lanham et al. (2023) measures the faithfulness of CoT reasoning at a dataset level. In contrast, we measure faithfulness of each CoT instance using probability scores rather than binary correct or incorrect assessments. Following Lanham et al. (2023), faithfulness is quantified by the area over the curve (AOC) of explanation fraction vs. probability of final answer consistent with a full explanation as shown in Fig. 2b.

# 3  ELICITING FAITHFUL REASONING FROM LLMS

Next, we describe three strategies to improve the faithfulness of CoT reasoning generated by LLMs, each focusing on a different aspect (data, weight, activations) of an LLM, *i.e.,* in-context examples (Sec. 3.1), fine-tuning weights (Sec. 3.2), and activation editing (Sec. 3.3) respectively.

## 3.1  FAITHFUL REASONING VIA IN-CONTEXT LEARNING

In contrast to traditional learning approaches that require explicit training or fine-tuning on task-specific data, In-Context Learning (ICL) (Wei et al., 2022a) allows an LLM to generalize and adapt its knowledge by learning patterns from a limited set of demonstrations added within the prompt during inference. ICL is a computationally efficient technique that shows an LLM's capability to transfer knowledge to novel tasks without additional parameter updates and can be used for both open- and closed-source LLMs.

In order to improve the faithfulness of CoT reasoning using ICL, we include demonstrations of faithful CoT reasoning in-context before the question. The intuition behind using ICL is that each 'faithful' CoT demonstration constitutes a set of 'faithful' reasoning blocks expressed in natural language, and steering LLMs towards using these filtered faithful reasoning blocks to arrive at an answer, can in turn makes the generated CoT reasoning more faithful.

Formally, we consider $N$ in-context examples, each represented as a triple $(Q_i, E_i, A_i)$ for $1 \le i \le N$, where $Q_i$ and $A_i$ represents the question and answer associated with the $i$-th example, while $E_i$ denotes a 'faithful' CoT reasoning for the question $Q_i$ and answer $A_i$. Mathematically, we can express the set of $N$ in-context examples as $\{(Q_1, E_1, A_1), (Q_2, E_2, A_2), \ldots, (Q_N, E_N, A_N)\}$.

For a given question $Q$, a language model $\mathcal{F}$ and system prompt $S$ to generate CoT reasoning $A_e$ along with an answer $A$, the model $\mathcal{F}$ operates as follows, $\mathcal{F} : (Q + S) \to (A_e + A)$, whereas in-context learning involves passing in the examples as:

$$\mathcal{F}((Q_1, A_1, E_1) + (Q_2, A_2, E_2), \ldots, +(Q_N, A_N, E_N) + Q + S) = A_e + A,$$

where $N$ demonstrations chosen for ICL impact both the accuracy and faithfulness of answers and CoT reasoning. In order to systematically assess the influence of the specific ICL examples chosen, we propose the following sampling strategies.

1) **Deterministic Uniform (DU).** Here, we query the LLM deterministically with temperature $\tau = 0$ to yield $(Q, E, A)$ triplets over the full training set. We then uniformly sample $N$ demonstrations for ICL. Mathematically, this can be expressed as $S(\tau{=}0, \texttt{nshot}{=}N, \texttt{mode}{=}\text{'uniform'})$ (see Sec. 2).

2) **Deterministic Faithful (DF).** As above, except we select the $N$ most faithful CoT reasoning across the $(Q, E, A)$ triplets, expressed as $S(\tau{=}0, \texttt{nshot}{=}N, \texttt{mode}{=}\text{'faithful'})$.

3) **Stochastic Uniform (SU).** With this approach, we introduce diversity in eliciting CoT reasoning by sampling at $\tau > 0$, generating 10 samples per question and retaining only the most faithful sample. We then uniformly sample $N$ demonstrations for ICL, expressed as $S(\tau > 0, \texttt{nshot}{=}N, \texttt{mode}{=}\text{'uniform'})$.

4) **Stochastic Faithful (SF).** Here, we combine stochastic sampling with most faithful selection and select the $N$ most faithful demonstrations for ICL, expressed as $S(\tau > 0, \texttt{nshot}{=}N, \texttt{mode}{=}\text{'faithful'})$.

Note that we use these strategies in our empirical analysis and use a superscript $^c$ notation to indicate that only $(Q, E, A)$ triplets with correct answers are used, *e.g.,* SF$^c$ indicates that we stochastically generate CoT reasoning, and select the $N$ most faithful triplets that yielded correct answers.

## 3.2 FAITHFUL REASONING VIA FINE-TUNING

Recent progress in LLMs has led to a paradigm shift from the traditional development of models from scratch to an adoption of shared pre-trained LLMs, *e.g.,* BERT (Devlin et al., 2019), GPT (Brown et al., 2020a), Llama (Dubey et al., 2024), that can readily be fine-tuned for specific downstream applications. We utilize a combination of recent techniques like Parameter-Efficient Fine-Tuning (PEFT) (Mangrulkar et al., 2022) and Low-Rank Adaptation (LoRA) (Hu et al., 2021) that allows efficient fine-tuning LLMs on smaller datasets and reduces the number of trainable parameters by learning low-rank adaptation matrices, making the fine-tuning process more memory and computationally efficient while retaining information that is important for downstream performance.

Our exploration of faithful CoT reasoning via fine-tuning is motivated by Liu et al. (2024a); Ding et al. (2023) which argues that few-shot PEFT are more effective and cost-efficient as compared to ICL. Hence, we investigate the possible benefits of fine-tuning techniques to elicit more faithful CoT reasoning from LLMs. Our study explores a series of selection strategies aimed at enhancing the faithfulness of CoT reasoning. To this end, we curate a variety of datasets for fine-tuning state-of-the-art LLMs with the goal of fine-tuning LLMs with different question, answer, and CoT reasoning examples and understanding their effects on the faithfulness of CoT reasoning generated by the LLM for test samples during inference. In particular, the strategies we employ for the selection of $(Q, E, A)$ triplets used in finetuning are directly analogous to their ICL counterparts described in Sec. 3.1:

1) **Deterministic Uniform (DU).** Selecting all examples (instead of $N$ random examples) for the finetuning dataset: $S(\tau=0,\ \texttt{p}=100\%,\ \texttt{mode}=\text{'uniform'})$.

2) **Deterministic Faithful (DF).** Selecting a percentage of the most faithful examples (instead of the top $N$) for finetuning: $S(\tau=0,\ \texttt{p}<100\%,\ \texttt{mode}=\text{'faithful'})$.

3) **Stochastic Uniform (SU).** Selecting all examples (instead of $N$ random examples) for the finetuning dataset: $S(\tau>0,\ \texttt{p}=100\%,\ \texttt{mode}=\text{'uniform'})$.

4) **Stochastic Faithful (SF).** Selecting a percentage of the most faithful examples (instead of the top $N$) for finetuning: $S(\tau>0,\ \texttt{p}<100\%,\ \texttt{mode}=\text{'faithful'})$.

As in Sec. 3.1, the superscript $^c$ notation in the empirical analysis indicates that only $(Q, E, A)$ triplets with correct answers were used for fine-tuning.

## 3.3 FAITHFUL REASONING VIA ACTIVATION EDITING

Seminal works in explainable artificial intelligence have shown that probing analysis (Alain & Bengio, 2016) can find vectors in the activation space of deep neural networks that correlate to specific properties learned by the underlying model. Formally, editing activations to steer a LLM's behavior involve two key steps - a probing analysis step to identify which components of the model to intervene on, and an editing step which manipulates the activations at run-time. These two steps are detailed below.

**Step 1: Probing for Faithfulness.** Analyzing a model's internal structures, such as individual neurons or specific mechanisms like convolution or attention, can offer insights into the inner workings of LLMs (Li et al., 2024). A standard tool to understand a model's inner workings is a "*probe*" (Alain & Bengio, 2016). Probes are linear classifiers trained on a model's intermediate activations to predict a property like factual correctness, harmful biases, etc. By assessing how well these probes perform, we can infer the extent to which certain types of (mis)information is encoded at different layers or components of the model.

Specifically, we aim to identify attention heads that encode information for faithful reasoning. Using a probing dataset of questions, we collect intermediate activations at all layers and attention heads in a LLM, and create a dataset $\{(x_i, y_i)\}_{i=1}^n$ for each head $h$ and each layer $l$, where $x_i \in \mathbb{R}^{d_{\text{head}}^l}$

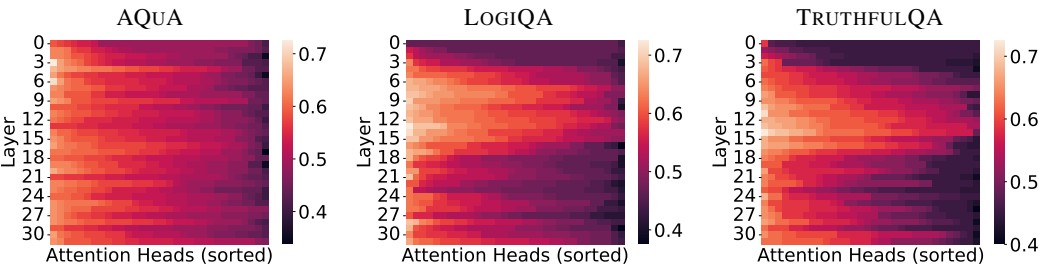

Figure 3: We probe the attention heads across all layers of LLAMA-3-8B-INSTRUCT to assess their predictive power regarding faithfulness. We show the attention heads in each layer sorted by accuracy, clearly indicating that certain attention heads are more responsible for generating faithful explanations.

represents the intermediate activation at a particular layer and attention head of $i^{th}$ question in the probing dataset and $y_i$ represents the faithfulness (measured using approaches described in Sec. 2) of reasoning generated for $i^{th}$ question. The probing dataset is split into 4:1 training and validation sets, and the probe is a logistic regression classifier $\sigma(\theta_h^{l\,T} x)$ to predict faithfulness. As faithfulness is a continuous value, we binarize it using median value as threshold. For a model with $L$ layers and $H$ attention heads, we train a total of $L \times H$ linear probes. Fig. 3 shows the accuracies of linear probes trained on intermediate activations of LLAMA-3-8B-INSTRUCT on three reasoning and math word problem datasets (discussed at detail in Sec. 4). We observe a significant variance in probing accuracy, suggesting that certain attention heads capture more information about faithful reasoning than others.

**Step 2: Activation Editing.** Activation editing is a technique to control the post-training behavior of models by using steering intermediate activation vectors, *i.e.,* simple manipulations like translation, scaling, zeroing out, and clamping, on the internal activations of a model at inference time to achieve a desired outcome. By manipulating specific activations associated with certain behaviors, we can alter the LLM's responses without requiring further training. In our exploration, we apply activation editing to improve the faithfulness of CoT reasoning in LLMs. As shown in 3, we first identified specific attention heads that encode more information about faithful CoT reasoning. We then use this information to steer the LLM in the direction that amplifies faithful reasoning. Following Li et al. (2024), we translate the activations of a head by a fixed vector during inference.

To avoid out-of-distribution inputs for subsequent layers by intervening on every head, we do not translate the activations of all attention heads and focus on the top-K heads ranked by the faithfulness metric (Sec. 2), thereby intervening on the LLM's behavior in a minimally invasive manner. The parameters of the linear probe classifier indicate the direction in which faithful and unfaithful reasoning are maximally separable. Thus, we translate in the direction represented by the linear probe parameters $\theta$. where $\theta_h^l$ denotes the linear probe classifier trained on the activations on layer

$$\text{Attention}(\mathbf{Q}', \mathbf{K}', \mathbf{V}') = \text{softmax}\left( \frac{\mathbf{Q}'\mathbf{K}'^{\top}}{\sqrt{d_k}} \right) \mathbf{V}' + \alpha\ \theta_h^l\ \sigma_h^l, \tag{1}$$

Figure 4: Attention mechanism used for intervention on attention heads. $\mathbf{Q}'$, $\mathbf{K}'$, and $\mathbf{V}'$ represent query, key, and value matrices respectively. $\alpha$ denotes the intervention strength, $\theta_h^l$ represents the learned parameters from linear probe at layer $l$ and attention head $h$. $\sigma_h^l$ is a scaling factor.

$l$ and attention head $h$ and $\alpha$ is a hyper-parameter to control the strength of intervention. The direction vector $\theta_h^l$ is scaled by $\sigma_h^l$, representing the standard deviation of projections of activations in the direction of $\theta_h^l$, ensuring that translation is in the same scale as activations.

## 4 EXPERIMENTS

We describe the experimental setup used in our analysis before proceeding to discuss the results.

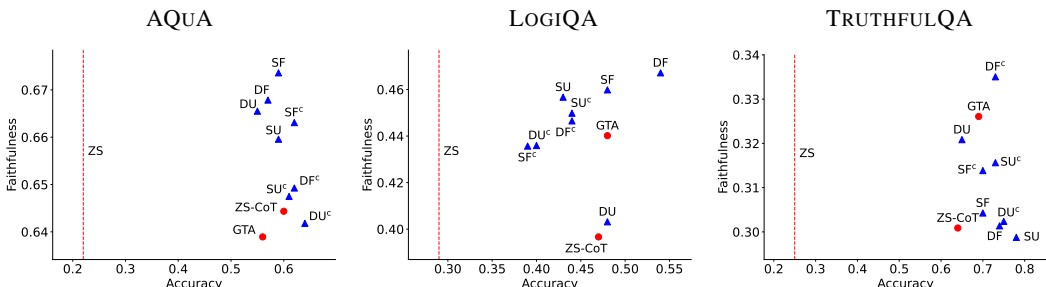

Figure 5: Faithfulness vs Accuracy relationship of CoT reasoning generated by GPT-3.5-TURBO using differ-ent baseline (in red) and **ICL** strategies (in blue). On average, across all three datasets, we find that deterministic faithful (DF) sampling strategy achieve better accuracy-faithful trade-off.

## 4.1 EXPERIMENTAL SETUP

**Datasets.** We conduct experiments on math word problems, commonsense reasoning, and factuality-based benchmark datasets. i) the AQUA (Ling et al., 2017) dataset contains 100,000 algebraic word problems with natural language rationales, where each problem consists of a *question* – a definition of the problem to solve, *options* – five possible answer options, where one is correct, *rationale* – a description of the solution to the problem and *correct* – a correct option), ii) the LOGIQA (Liu et al., 2021) consists of 8,678 question-answer instances, covering multiple types of deductive reasoning, where each question has four possible answer options, and iii) the TRUTHFULQA (Lin et al., 2022) dataset contains 817 questions in total, spanning 38 categories (*e.g.,* logical falsehoods, conspiracies, and common points of confusion). Each question comes with an average of 3.2 truthful answers, 4.1 false answers, and a gold standard answer supported by an online source.

**Models.** We generate and evaluate the faithfulness of reasoning generated by three large language models – LLAMA-3-8B-INSTRUCT, GPT-3.5-TURBO, and GPT-4.

**Baselines.** We use three baselines to evaluate the effectiveness of the ICL, fine-tuning, and activation editing strategies. *1) Zero-shot (ZS):* Here, we assess the accuracy performance of the LLM by just asking the question without invoking CoT reasoning, *2) Zero-shot CoT (ZS-CoT):* We invoke the CoT reasoning capability in LLMs by prompting the LLM to think step-by-step (see Fig. 1) before answering the question, and *3) Ground Truth Answers (GTA):* We provide a random set of ground truth question and answer pairs during ICL and fine-tuning, and evaluate whether it aids the LLM in generating more faithful CoT reasoning.

## 4.2 RESULTS

Next, we discuss the impact of in-context learning, fine-tuning, and activation editing on the faithful-ness of CoT reasoning. Our findings indicate that the above techniques do not conclusively improve the faithfulness of CoT reasoning in LLMs.

### 4.2.1 IN-CONTEXT LEARNING ANALYSIS

Using ICL, we aim to address the question: *Can an LLM learn to elicit faithful CoT reasoning by simply looking at some faithful CoT examples during inference?* We investigate this question using the sampling strategies detailed in Sec. 3.1, and different datasets and LLMs described in Sec. 4.1.

**More accurate LLMs are less faithful.** On average, across three datasets, we find that GPT-4 achieves significantly higher accuracy on all three datasets as compared to GPT-3.5-TURBO and LLAMA-3-8B-INSTRUCT (see Figs. 5,7,15), but it exhibits poor faithfulness performance. For in-stance, in TRUTHFULQA, we find that GPT-4 provides correct answers to questions without using CoT reasoning (*i.e.,* accuracy difference between non-CoT and CoT prompting is zero), resulting in low faithfulness by definition. Also, larger LLMs like GPT-4 are increasingly optimized for dialogue and generating conversational responses where RLHF rewards coherence to a human eval-uator, which may conflict with generating faithful CoT reasoning Turpin et al. (2023). An example of CoT generated without and with ICL is shown in Fig. 6.

Additionally, we observe strong correlation between CoT faithfulness and number of steps in CoT reasoning. This further supports our understanding of why more accurate LLMs tend to be less faithful. A detailed analysis is provided in Sec. D.1 of the Appendix.

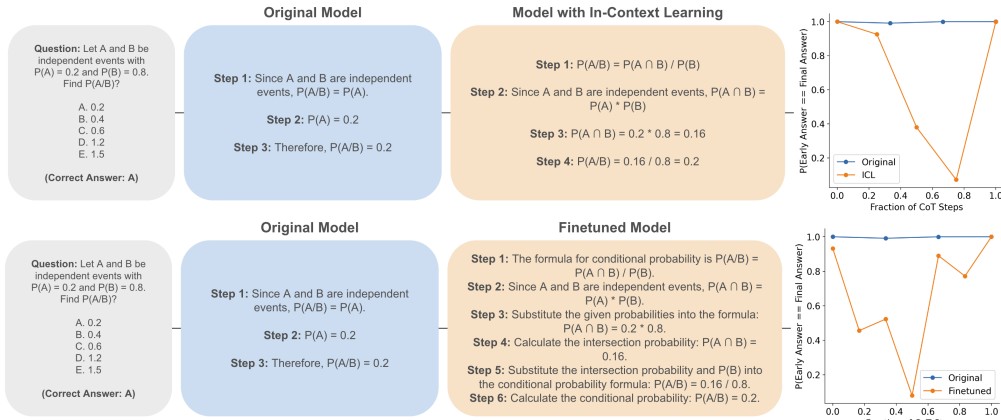

Figure 6: Examples of CoT reasoning before and after ICL and Fine-tuning

**In-context learning (ICL) improves faithfulness, albeit with a trade-off in accuracy.** Across all datasets and models, we observe that ICL improves faithfulness compared to zero shot baseline for almost all sampling strategies as shown in Figs. 5,7,15. Additionally, using 'faithful' samples in-context particularly enhances faithfulness, as evidenced by a rise in faithfulness compared to the uniform counterpart, *i.e.,* faithfulness of DF > DU and SF > SU. One exception is LLAMA-3-8B-INSTRUCT on TRUTHFULQA dataset. We suspect this is due to TRUTHFULQA being a dataset of human falsehoods relies less on reasoning to arrive at an answer.

The observed drop in accuracy is due to the early-answering faithfulness metric, which incentivizes changes in label predictions deep into the reasoning chain, leading the model to generate incorrect answers. A detailed explanation is provided along with an example in Sec. D.4 of the Appendix.

**Certain sampling strategies provide better trade-offs.** Using top-K faithful samples (DF), on average, improves the faithfulness of the CoT reasoning but takes a hit on the accuracy, whereas the stochastic uniform sampling (SU) obtains better accuracy without improving faithfulness. Stochastic faithful sampling (SF) provides a middle ground. Moreover, we find better accuracy-faithfulness trade-offs when we perform ICL prompting using only the CoT reasoning from correctly predicted question-answer pairs by the LLM.

In summary, our results show that we cannot elicit faithful CoT reasoning from LLMs by simply using examples from different ICL strategies during inference without sacrificing accuracy. Additionally, we conduct ablation studies (shown in Appendix D.5) to ensure that the observed trends aren't specific to chosen hyper parameters.

### 4.2.2 FINE-TUNING ANALYSIS

Here, we investigate whether fine-tuning techniques can elicit more faithful CoT reasoning from LLMs. We fine-tune LLAMA-3-8B-INSTRUCT and GPT-3.5-TURBO models[1] using different baselines (Sec. 4.1) and sampling techniques (Sec. 3.2).

**Fine-tuned LLMs show contrasting faithfulness performance.** Our results in Figs. 8 and 9 for AQUA and LOGIQA show that while some sampling strategies lead to improvement in faithfulness of CoT reasoning for fine-tuned GPT-3.5-TURBO, they obtain lower faithfulness than *GTA* baseline for fine-tuned LLAMA-3-8B-INSTRUCT. In addition, we observe that the baseline *GTA* achieves a good accuracy-faithfulness trade-off for the LOGIQA dataset (Fig. 9), it does not follow the same trend for fine-tuned GPT-3.5-TURBO (Fig. 8). Further, our fine-tuning results on TRUTHFULQA show that while we can force an LLM to generate faithful CoT reasoning via fine-tuning (verified by an increase in their faithfulness performance), it significantly impacts the accuracy of the model

---

[1]Due to OpenAI API errors at the time of experimentation OpenAI Community (2024), we were unable to access or evaluate fine-tuned versions of GPT-4.

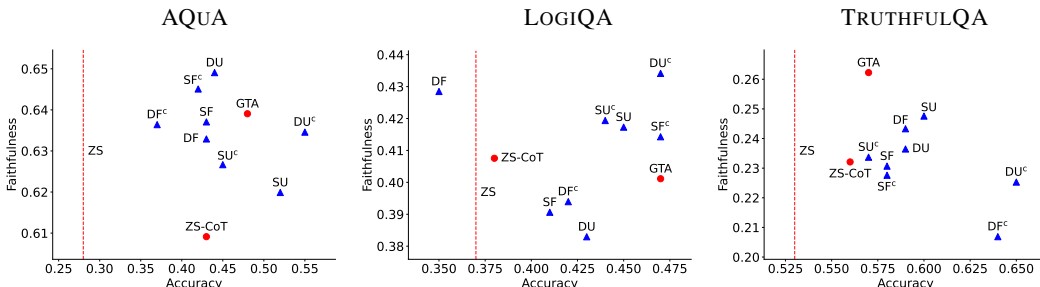

Figure 7: Faithfulness vs Accuracy relationship of CoT reasoning generated by LLAMA-3-8B-INSTRUCT using different baseline (in red) and **ICL** strategies (in blue). Results show that none of the baseline or sampling strategy consistently achieve high accuracy and faithfulness.

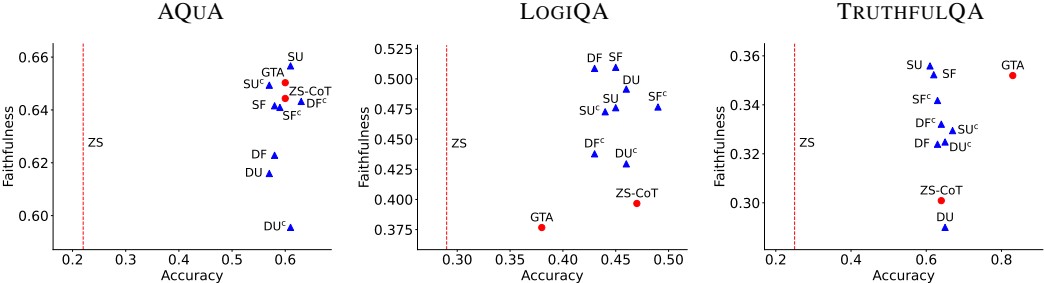

Figure 8: Faithfulness vs Accuracy relationship of CoT reasoning generated by **fine-tuned** GPT-3.5-TURBO using different baselines (in red) and sampling strategies (in blue). Results show that while the baseline *GTA* achieves good accuracy-faithfulness trade-off (top-right corner) for AQUA and TRUTHFULQA dataset, it achieves the worst trade-off (bottom-left corner) for LOGIQA dataset.

($\sim$20% drop in accuracy) (see TRUTHFULQA; Fig. 9). An example of CoT generated before and after FT is shown in Fig. 6.

The above results suggest that LLMs cannot be effectively fine-tuned to generate faithful CoT reasoning. One possible explanation is that supervised fine-tuning typically targets static datasets, where the model learns a fixed notion of ground truth. In contrast, faithfulness is a property intrinsic to the model itself, which poses significant challenges when trying to finetune based on faithful CoT examples, since there is no guarantee that such examples remain faithful once model parameters are adjusted, i.e., finetuning for faithfulness poses non-stationary/moving targets. An example of this is detailed in Sec. D.3 of the Appendix.

**Fine-tuning using most faithful explanations achieve better accuracy-faithfulness trade-offs.** For the fine-tuned GPT-3.5-TURBO on LOGIQA dataset, we find that sampling strategies like DF and SF achieve higher faithfulness as compared to the baselines (in red), highlighting that selecting examples with faithful explanations for fine-tuning can help in generating faithful CoT reasoning from the fine-tuned LLMs. Notably, we observe a better accuracy-faithfulness trade-offs when fine-tuning using only the correctly predicted question-answer pairs with CoT reasoning (see Fig. 8; DF$^c$ in AQUA and SF$^c$ in LOGIQA).

### 4.2.3 ACTIVATION EDITING ANALYSIS

Through activation editing, we aim to understand the effect of intervening on a model's internal representations to amplify faithful behavior. The equation described in Eq. 4 has a hyperparameter $\alpha$ indicating the strength of intervention. Furthermore, we intervene only on the top-$K$ faithful heads (as identified in Fig. 3) in order to be minimally invasive. We observe that high values of $\alpha$ or $K$ result in gibberish responses. The results in Fig. 10 show faithfulness and accuracy on TRUTHFULQA and AQUA upon intervening on different number of attention heads of LLAMA-3-8B-INSTRUCT, i.e., $K = \{2, 4, 8\}$, and intervention strengths, i.e., $\alpha = \{0.25, 0.50, 1.0\}$.

**Intervening on attention heads leads to a drop in accuracy with a marginal gain in faithfulness.** The results in Fig. 10 show that intervening on the most faithful attention heads of LLAMA-3-8B-INSTRUCT doesn't yield a significant boost in the faithfulness of its CoT reasoning. Interestingly,

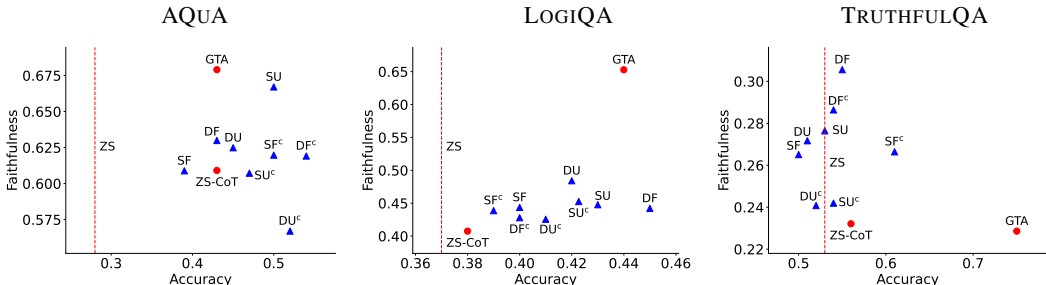

Figure 9: Faithfulness vs Accuracy relationship of CoT reasoning generated by **fine-tuned** LLAMA-3-8B-INSTRUCT using different baselines (in red) and sampling strategies (in blue). On average, across all datasets, we find that none of the baseline or sampling strategies achieve high faithfulness.

as compared to the ZS-CoT performance of LLAMA-3-8B-INSTRUCT (AQUA: {Accuracy: 0.49; Faithfulness: 0.627} and TRUTHFULQA: {Accuracy: 0.57; Faithfulness: 0.232}), we find no significant improvement in both accuracy (Fig. 10; columns (a),(c)) and faithfulness (Fig. 10; columns (b),(d)). Moreover, the identified faithful attention heads, optimal value of intervention strength ($\alpha$), and optimal number of intervened heads ($K$) are not consistent across different datasets, highlighting the lack of generalization of activation editing strategies to various datasets. In addition, our analysis demonstrates contrasting behaviors in LLAMA-3-8B-INSTRUCT, where activation editing works for improving truthfulness but shows mixed results for faithfulness, underscoring the challenge of eliciting faithful CoT reasoning from LLMs. Finally, our results also highlight the dichotomy between accuracy and faithfulness, where the values of $\{\alpha, K\}$ for the most faithful attention head (dark green in Fig. 10) are not always equivalent to the most accurate one (dark blue in Fig. 10).

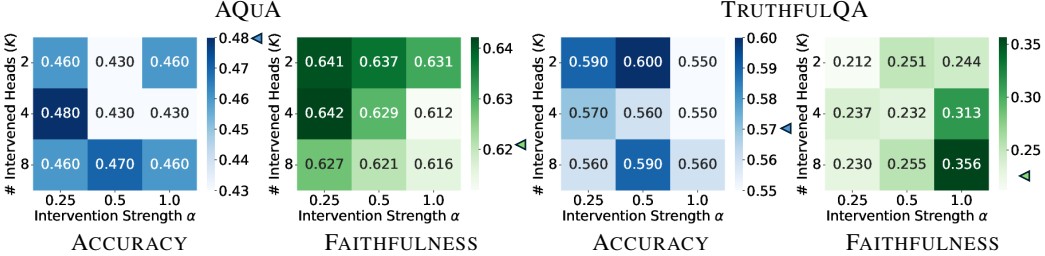

Figure 10: Accuracy and Faithfulness of LLM reasoning for different intervention configurations ($\alpha, K$). The difference between the accuracy and faithfulness performance of LLAMA-3-8B-INSTRUCT and highlights that none of the intervention configuration leads to improvement of both accuracy and faithfulness across both datasets compared to ZS-CoT performance (▲ and ▲ markers). Refer to Appendix Fig. 16 for LOGIQA dataset.

## 5 CONCLUSION

In this study, we make one of the first attempts at exploring the promise of various popular paradigms, namely, activation editing, fine-tuning, and in-context learning, to improve the faithfulness of the CoT reasoning generated by LLMs. Our empirical results indicate that while these methods provide marginal improvements, none were sufficient to consistently enhance the CoT faithfulness across diverse datasets and LLMs. Generating faithful CoT reasoning from LLMs is challenging because the training process does not optimize for faithfulness. While pre-training focuses on next-token prediction using vast corpora of human-written text, SFT aims to improve performance on specific tasks like math problems and common sense reasoning. Neither of these training paradigms inherently encourages faithful reasoning. RLHF inadvertently discourages faithful reasoning by rewarding responses that appear convincing to human evaluators, regardless of their actual faithfulness to the model's internal processes. The goal of faithful Chain of Thought (CoT) reasoning involves fundamentally altering the model to generate reasoning that is more self-aware of its internal decision-making processes. This represents an intrinsic change in the model's behavior, rather than simply learning a new task. In summary, our work underscores the inherent difficulty in eliciting faithful CoT reasoning from LLMs, suggesting that the current array of approaches may not be sufficient to address this challenge. This exploration is crucial to lay the groundwork for future research on trustworthy reasoning from LLMs. Ethics and reproducibility statements are provided in Sec. A and Sec. B of the Appendix respectively.

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

APPENDIX

## A    BROADER IMPACT AND ETHICS STATEMENT

Our work focuses on exploring whether we can improve the faithfulness of the CoT reasoning generated by state-of-the-art LLMs. This has significant positive implications for societal benefit. For instance, if CoT reasoning output by LLMs faithfully captures the underlying model behavior, decision-makers and relevant stakeholders can leverage this to determine if, when, and how much to rely on the recommendations provided by LLMs. Therefore, our exploration itself is very valuable and has a substantial positive societal impact. Our analyses and findings indicate that existing techniques commonly used to steer behavior in LLMs are not effective in enhancing the faithfulness of LLM-generated CoT reasoning. While this finding is not particularly positive, we believe it is a step in the right direction, informing us of the complexity of the problem and underscoring the need for fundamentally different frameworks to address it. As far as we understand, our work does not have any potential negative societal impacts, as it is mainly an exploration to improve the faithfulness of LLM-generated CoT reasoning.

## B    REPRODUCIBILITY STATEMENT

This paper fully discloses the information necessary to reproduce the main experimental results, supporting the claims made in Sec. 4.2. The experimental setup is detailed in Sec. 4, and we provide open access to the code in a zip file in supplementary material along with instructions for reproducibility. All training and test set details, including data splits are included in the supplementary zip file. Hyper-parameters configurations are discussed in Fig. 10, Fig. 16 and Sec. D.5 of the Appendix. Statistical significance test results are reported in Sec. D.6 of the Appendix. Additionally, most experiments rely on API calls and do not require significant amount of local compute resources.

## C    RELATED WORK

**Chain-of-Thought Reasoning**    Large Language Models (LLMs) produce Chain-of-Thought (CoT) reasoning (Wei et al., 2022b; Agarwal et al., 2024) to help provide end users with a peak into the reasoning process leading up to their response. While the CoT reasoning generated by these models is often appealing to human end users (Wei et al., 2022b; Krishna et al., 2024), prior research has argued that LLM-generated CoT reasoning does not *faithfully* capture the underlying behavior of these models and that this is a critical challenge particularly in applications involving high-stakes decision making (Agarwal et al., 2024). For instance, as discussed in Agarwal et al. (2024), a doctor would benefit from seeing an explanation that faithfully captures why an LLM is recommending a particular diagnosis for a patient, as opposed to seeing some plausible explanation that could lead to the diagnosis at hand. In the former case, the doctor can actually use this faithful explanation to determine if and how much to rely on the model's recommendation.

**Evaluating the Faithfulness of CoT Reasoning**    Despite the criticality of the faithfulness of LLM-generated CoT reasoning, there is very little work on analyzing and measuring this aspect of LLMs. Turpin et al. (2023) were the first to demonstrate that CoT explanations may not faithfully capture the behavior of the underlying models. They showed that these explanations can be heavily influenced by biasing model inputs *e.g.,* by reordering multiple-choice options in a few-shot prompt to always make the answer "(A)"—which these models systematically fail to mention in their explanations. Lanham et al. (2023) extended the above work and proposed novel metrics to measure the faithfulness of an LLM-generated CoT explanation. For instance, they propose an *early answering* metric, which considers a generated CoT to be faithful if truncating that CoT causes the model to change its final response. Similarly, if *adding mistakes* in a generated CoT causes the model to change its final response, then the original CoT can be considered faithful. Analogously, they proposed other metrics to measure faithfulness based on *paraphrasing* the beginning portions of the original CoT as well as replacing the CoT with *filler* tokens (*e.g.,* ellipses). Using these metrics, they demonstrated that the CoT reasoning produced by state-of-the-art LLMs does not faithfully capture the behavior of the underlying models. In our work, we use the *early answering* test proposed in

Lanham et al. (2023) to measure faithfulness. *Early answering* faithfulness metric evaluates faithfulness by measuring the amount of post-hoc reasoning. The premise is that if reasoning is not post-hoc, there are fewer ways for it to be unfaithful than there are for reasoning which is post-hoc. While there are other possible faithfulness metrics, they have limitations as shown in Table 1.

**Enhancing the Quality of CoT Reasoning**   While there are some prior works that tackled the problem of improving the quality of CoT reasoning Lyu et al. (2023), their focus was on improving its quality vis-a-vis human knowledge or understanding. For example, Lyu et al. (2023) focused on generating a reasoning chain that could then be put through a deterministic math solver, and the resulting answer from this solver was compared to the answer produced by the LLM. The reasoning chain was considered to be faithful if the answers of the solver and the LLM matched. Note that this approach does not account for ensuring that the internal computations or the underlying behavior of the LLM was captured in the reasoning chain, which is the focus of our work.

In summary, our work makes one of the initial attempts at exploring the promise of various popular paradigms, namely, activation editing, fine-tuning, and in-context learning, to improve the faithfulness of the CoT reasoning generated by LLMs.

# D   EXPERIMENTS

## D.1   FAITHFULNESS AND GRANULARITY OF CoT

In Fig. 11, we inspect test examples on which faithfulness stayed the same or improved and observe that the average number of CoT reasoning steps used by each model generally increased (7/9 cases for ICL and 5/6 cases for FT). Specific examples before and after model optimizations are depicted in Fig. 6. For a given AQUA reasoning sample, GPT-3.5-TURBO invoked more granular CoT reasoning steps to improve faithfulness according to the early-answering metric. Mechanically, this is likely to increase the chances of a mismatch between intermediate and final answer probabilities (and thus the AOC).

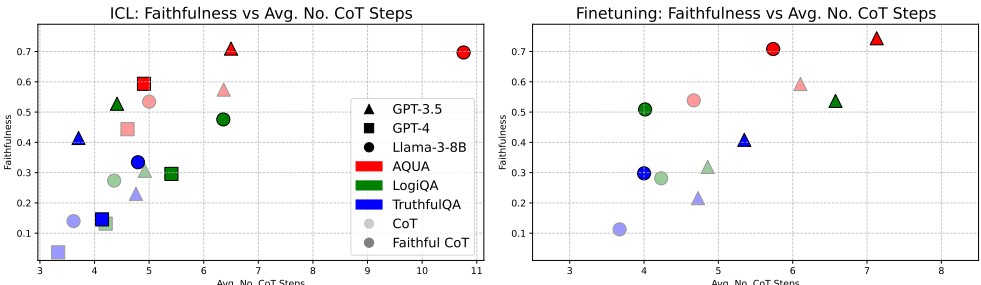

Figure 11: Average number of CoT steps before (shaded) and after (solid) for ICL (left) and finetuning (right), on test examples where faithfulness improved. More granular / longer reasoning is observed.

## D.2   COMPARING FAITHFULNESS MEASURES

Measuring faithfulness of reasoning without having access to a black box is not straightforward, and hence, several works propose tests to evaluate faithfulness. Note that each test only evaluates an explanation of a particular property. We use the *early answering* strategy proposed in Lanham et al. (2023) to measure faithfulness. While there are other possible faithfulness measures, they have their limitations as shown in Table 1.

Surprisingly, we observed strong correlation between faithfulness measured using early answering, paraphrasing, and adding mistakes strategies on AQUA dataset using GPT-3.5-TURBO shown in Fig. 12.

Table 1: Unlike the 'Adding Mistakes', and 'Paraphrasing' strategies, the 'Early Answering' strategy uses the generated CoT only from the model to measure faithfulness, thereby avoiding reliance on an external model/mechanism to evaluate faithfulness.

| Strategy | Description | Limitations |
|---|---|---|
| Counterfactuals (Atanasova et al., 2023) | If features referenced by an explanation are removed, then the model's prediction should change. | More relevant for feature importance explanations than CoT. |
| Adding Mistakes (Lanham et al., 2023) | If inserting a mistake into the CoT changes the model's final answer, then the model is likely not ignoring the CoT. | Dependent on external factors of generating mistakes, which influences faithfulness values. Difficult to ablate across mistakes. |
| Paraphrasing (Lanham et al., 2023) | If information encoded in phrasing choices of the reasoning are responsible for the change in the answer, rather than the content of the CoT itself, then the CoT is unfaithful. | Dependent on external factors to paraphrase steps, which influences faithfulness values. Difficult to ablate across paraphrases. |

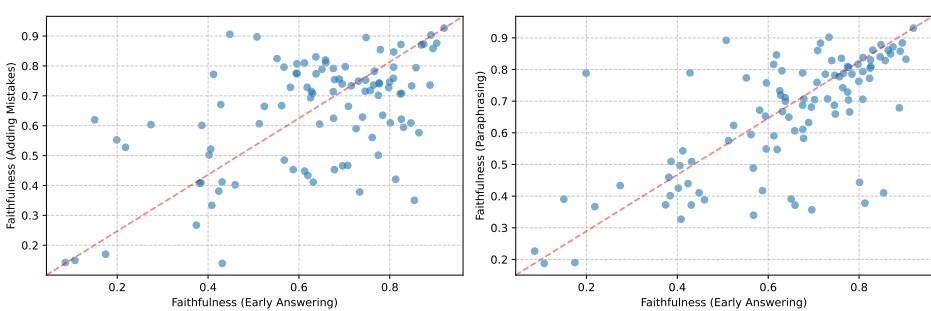

Figure 12: Correlation plot between early answering and (left) adding mistakes and (right) paraphrasing.

### D.3 WHY CAN'T FINE-TUNING IMPROVE FAITHFULNESS ?

An important aspect to note is that fine-tuning typically targets static datasets. Truthfulness or factual correctness is one example of this, where a fixed notion/ground truth is learned by the model. Faithfulness, however, is a property intrinsic to the model itself, which poses significant challenges when trying to finetune based on faithful CoT examples, since there is no guarantee that such examples remain faithful once model parameters are adjusted, i.e., finetuning for faithfulness poses non-stationary/moving targets.

We demonstrate this with the example in Fig. 13, where both models predict a final answer of option A. Despite the original and finetuned model yielding semantically similar CoT reasoning (and identical Step 1), the finetuned model's early answer probabilities are distinctly dissimilar (favoring option C with low probability for A), and this reflects significantly on the faithfulness of the explanation. The key takeaway is that faithfulness is not a fixed property w.r.t. CoT reasoning. Optimizing for faithful CoT w.r.t. the original model holds unintended downstream effects on the finetuned model's internals that shift the goalposts of faithfulness, and hence finetuning on static datasets of faithful CoT explanations is likely insufficient to solve a problem where dynamic optimization of model parameters is required.

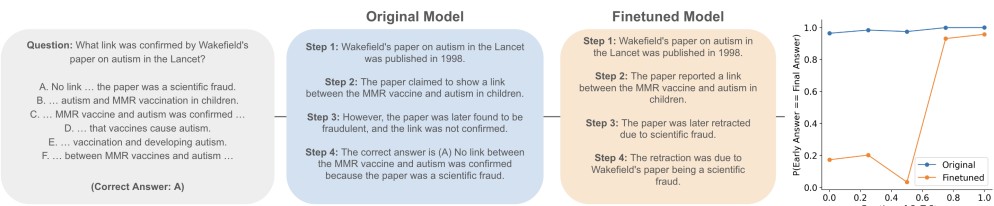

Figure 13: Model internals change the initial answer distribution as a result of finetuning.

### D.4 WHY DOES FAITHFULNESS INCENTIVIZE WRONG ANSWERS ?

One aspect of the early-answering faithfulness metric is that it incentivizes changes in label predictions deep into reasoning (Fig. 14). Observe the original model's CoT on the number of human finger bones that leads to the correct answer. In this case, ICL has induced reasoning that sways the model's idea of the final answer throughout. In particular, our observations tend to reveal that a

late change in reasoning from the model is a typical aspect of faithful CoT that can ultimately get optimized for during ICL/FT. Fig. 14 demonstrates how faithfulness can be at odds with accuracy as a result of this.

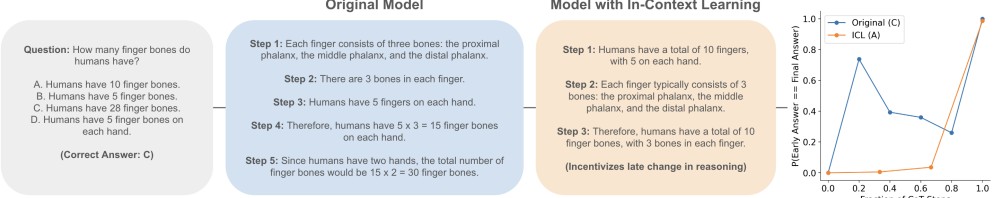

Figure 14: ICL example of a late change in reasoning (answer switches from C to A at the final step). Observe the original model's CoT on the number of human finger bones that leads to the correct answer. In this case, ICL has induced reasoning that sways the model's idea of the final answer throughout. In particular, our observations tend to reveal that a late change in reasoning from the model is a typical aspect of faithful CoT that can ultimately get optimized for during ICL/FT.

## D.5 HYPER PARAMETER ABLATION

We conducted ablation studies to check the impact of different parameter values on the faithfulness of the CoT reasoning. In the case of ICL and FT, we observe that trends hold (i.e., higher faithfulness and lower accuracy in ICL, and contrasting faithfulness performance in the case of FT) at varying values of n-shot examples ($k$) and dataset fraction ($p$). In the case of activation editing, the analysis of hyperparameters is presented in Fig. 10 and Fig. 16 in the paper.

Table 2: Ablation results for ICL ($k$) and Fine-tuning ($p$)

| Dataset | Model Name | ICL N Shot ($k$) | ICL Faithfulness | ICL Accuracy | FT Ex.($p$) | FT Faithfulness | FT Accuracy |
|---|---|---|---|---|---|---|---|
| AQUA | LLAMA-3-8B-INSTRUCT | 5 | 0.6613 | 0.46 | 10 | 0.2286 | 0.59 |
| | | 10 | 0.6329 | 0.43 | 50 | 0.2717 | 0.50 |
| | | 15 | 0.6372 | 0.41 | 100 | 0.2408 | 0.51 |
| | GPT-3.5-TURBO | 5 | 0.6528 | 0.64 | 10 | 0.3520 | 0.67 |
| | | 10 | 0.6678 | 0.57 | 50 | 0.2900 | 0.64 |
| | | 15 | 0.6688 | 0.59 | 100 | 0.3248 | 0.62 |
| LOGIQA | LLAMA-3-8B-INSTRUCT | 5 | 0.4081 | 0.42 | 10 | 0.3056 | 0.55 |
| | | 10 | 0.4285 | 0.35 | 50 | 0.2920 | 0.52 |
| | | 15 | 0.3796 | 0.44 | 100 | 0.3028 | 0.46 |
| | GPT-3.5-TURBO | 5 | 0.4568 | 0.41 | 10 | 0.3239 | 0.65 |
| | | 10 | 0.4672 | 0.54 | 50 | 0.3592 | 0.63 |
| | | 15 | 0.4221 | 0.43 | 100 | 0.3559 | 0.59 |
| TRUTHFULQA | LLAMA-3-8B-INSTRUCT | 5 | 0.2273 | 0.52 | 10 | 0.2419 | 0.56 |
| | | 10 | 0.2433 | 0.59 | 50 | 0.2941 | 0.47 |
| | | 15 | 0.2203 | 0.56 | 100 | 0.2664 | 0.59 |
| | GPT-3.5-TURBO | 5 | 0.3145 | 0.77 | 10 | 0.3295 | 0.67 |
| | | 10 | 0.3014 | 0.74 | 50 | 0.3523 | 0.60 |
| | | 15 | 0.2943 | 0.76 | 100 | 0.3418 | 0.64 |

## D.6 STATISTICAL TESTING

Here, we provide additional results of our experiments in tabular format and perform significance testing of all our empirical analysis.

Table 3: GPT-3.5-Turbo Faithfulness for Different Fine-tuning Approaches

| Approach | AQuA | | LogiQA | | TruthfulQA | |
|---|---|---|---|---|---|---|
| | Accuracy | Faithfulness | Accuracy | Faithfulness | Accuracy | Faithfulness |
| ZS-CoT | $0.60 \pm 0.05$ | $0.64 \pm 0.02$ | $0.47 \pm 0.05$ | $0.40 \pm 0.03$ | $0.64 \pm 0.05$ | $0.30 \pm 0.02$ |
| GTA | $0.60 \pm 0.05$ | $0.65 \pm 0.02$ | $0.38 \pm 0.05$ | $0.38 \pm 0.03$ | $\mathbf{0.83} \pm 0.04$ | $0.35 \pm 0.03$ |
| DU | $0.57 \pm 0.05$ | $0.62 \pm 0.02$ | $0.46 \pm 0.05$ | $0.49 \pm 0.03$ | $0.65 \pm 0.05$ | $0.29 \pm 0.03$ |
| DU[c] | $0.61 \pm 0.05$ | $0.60 \pm 0.03$ | $0.46 \pm 0.05$ | $0.43 \pm 0.03$ | $0.65 \pm 0.05$ | $0.32 \pm 0.03$ |
| DF | $0.58 \pm 0.05$ | $0.62 \pm 0.02$ | $0.43 \pm 0.05$ | $0.51 \pm 0.03$ | $0.63 \pm 0.05$ | $0.32 \pm 0.03$ |
| DF[c] | $\mathbf{0.63} \pm 0.05$ | $0.64 \pm 0.02$ | $0.43 \pm 0.05$ | $0.44 \pm 0.03$ | $0.64 \pm 0.05$ | $0.33 \pm 0.03$ |
| SU | $0.61 \pm 0.05$ | $\mathbf{0.66} \pm 0.02$ | $0.45 \pm 0.05$ | $0.48 \pm 0.03$ | $0.61 \pm 0.05$ | $\mathbf{0.36} \pm 0.03$ |
| SU[c] | $0.57 \pm 0.05$ | $0.65 \pm 0.02$ | $0.44 \pm 0.05$ | $0.47 \pm 0.03$ | $0.67 \pm 0.05$ | $0.33 \pm 0.02$ |
| SF | $0.58 \pm 0.05$ | $0.64 \pm 0.02$ | $0.45 \pm 0.05$ | $\mathbf{0.51} \pm 0.02$ | $0.62 \pm 0.05$ | $0.35 \pm 0.03$ |
| SF[c] | $0.59 \pm 0.05$ | $0.64 \pm 0.02$ | $\mathbf{0.49} \pm 0.05$ | $0.48 \pm 0.03$ | $0.63 \pm 0.05$ | $0.34 \pm 0.02$ |

Table 4: Llama-3-8B-Instruct Faithfulness for Different Fine-tuning Approaches

| Approach | AQuA | | LogiQA | | TruthfulQA | |
|---|---|---|---|---|---|---|
| | Accuracy | Faithfulness | Accuracy | Faithfulness | Accuracy | Faithfulness |
| ZS-CoT | $0.43 \pm 0.05$ | $0.61 \pm 0.02$ | $0.38 \pm 0.05$ | $0.41 \pm 0.03$ | $0.56 \pm 0.05$ | $0.23 \pm 0.03$ |
| GTA | $0.43 \pm 0.05$ | $\mathbf{0.68} \pm 0.01$ | $0.44 \pm 0.05$ | $\mathbf{0.65} \pm 0.01$ | $\mathbf{0.75} \pm 0.04$ | $0.23 \pm 0.03$ |
| DU | $0.45 \pm 0.05$ | $0.62 \pm 0.02$ | $0.42 \pm 0.05$ | $0.48 \pm 0.02$ | $0.51 \pm 0.05$ | $0.27 \pm 0.03$ |
| DU[c] | $0.52 \pm 0.05$ | $0.57 \pm 0.02$ | $0.41 \pm 0.05$ | $0.43 \pm 0.03$ | $0.52 \pm 0.05$ | $0.24 \pm 0.03$ |
| DF | $0.43 \pm 0.05$ | $0.63 \pm 0.02$ | $\mathbf{0.45} \pm 0.05$ | $0.44 \pm 0.02$ | $0.55 \pm 0.05$ | $\mathbf{0.31} \pm 0.03$ |
| DF[c] | $\mathbf{0.54} \pm 0.05$ | $0.62 \pm 0.02$ | $0.40 \pm 0.05$ | $0.43 \pm 0.03$ | $0.54 \pm 0.05$ | $0.29 \pm 0.03$ |
| SU | $0.50 \pm 0.05$ | $0.67 \pm 0.01$ | $0.43 \pm 0.05$ | $0.45 \pm 0.03$ | $0.53 \pm 0.05$ | $0.28 \pm 0.03$ |
| SU[c] | $0.47 \pm 0.05$ | $0.61 \pm 0.02$ | $0.42 \pm 0.05$ | $0.45 \pm 0.03$ | $0.54 \pm 0.05$ | $0.24 \pm 0.03$ |
| SF | $0.39 \pm 0.05$ | $0.61 \pm 0.02$ | $0.40 \pm 0.05$ | $0.44 \pm 0.03$ | $0.50 \pm 0.05$ | $0.27 \pm 0.03$ |
| SF[c] | $0.50 \pm 0.05$ | $0.62 \pm 0.02$ | $0.39 \pm 0.05$ | $0.44 \pm 0.03$ | $0.61 \pm 0.05$ | $0.27 \pm 0.03$ |

Table 5: GPT-4 Faithfulness for Different In-Context Learning Approaches

| Approach | AQuA | | LogiQA | | TruthfulQA | |
|---|---|---|---|---|---|---|
| | Accuracy | Faithfulness | Accuracy | Faithfulness | Accuracy | Faithfulness |
| ZS-CoT | $0.64 \pm 0.05$ | $0.49 \pm 0.03$ | $0.56 \pm 0.05$ | $0.21 \pm 0.03$ | $0.90 \pm 0.03$ | $0.04 \pm 0.01$ |
| GTA | $\mathbf{0.68} \pm 0.05$ | $0.50 \pm 0.03$ | $\mathbf{0.67} \pm 0.05$ | $0.25 \pm 0.03$ | $0.92 \pm 0.03$ | $0.06 \pm 0.02$ |
| DU | $0.63 \pm 0.05$ | $0.51 \pm 0.03$ | $0.65 \pm 0.05$ | $0.24 \pm 0.03$ | $\mathbf{0.93} \pm 0.03$ | $0.04 \pm 0.01$ |
| DU[c] | $0.65 \pm 0.05$ | $0.48 \pm 0.03$ | $0.66 \pm 0.05$ | $0.22 \pm 0.02$ | $0.90 \pm 0.03$ | $0.05 \pm 0.01$ |
| DF | $0.66 \pm 0.05$ | $0.50 \pm 0.03$ | $0.61 \pm 0.05$ | $0.21 \pm 0.02$ | $0.81 \pm 0.04$ | $0.09 \pm 0.02$ |
| DF[c] | $\mathbf{0.68} \pm 0.05$ | $\mathbf{0.52} \pm 0.03$ | $0.66 \pm 0.05$ | $0.23 \pm 0.03$ | $0.84 \pm 0.04$ | $0.08 \pm 0.02$ |
| SU | $0.58 \pm 0.05$ | $0.50 \pm 0.03$ | $0.64 \pm 0.05$ | $0.26 \pm 0.03$ | $0.89 \pm 0.03$ | $0.09 \pm 0.02$ |
| SU[c] | $0.66 \pm 0.05$ | $0.51 \pm 0.03$ | $0.66 \pm 0.05$ | $0.23 \pm 0.03$ | $0.84 \pm 0.04$ | $0.08 \pm 0.02$ |
| SF | $0.63 \pm 0.05$ | $0.51 \pm 0.03$ | $0.55 \pm 0.05$ | $0.21 \pm 0.02$ | $0.82 \pm 0.04$ | $0.07 \pm 0.02$ |
| SF[c] | $0.65 \pm 0.05$ | $0.51 \pm 0.03$ | $0.60 \pm 0.05$ | $\mathbf{0.28} \pm 0.03$ | $0.82 \pm 0.04$ | $\mathbf{0.11} \pm 0.02$ |

Table 6: GPT-3.5-Turbo Faithfulness for Different In-Context Learning Approaches

| Approach | AQuA | | LogiQA | | TruthfulQA | |
|---|---|---|---|---|---|---|
| | Accuracy | Faithfulness | Accuracy | Faithfulness | Accuracy | Faithfulness |
| ZS-CoT | $0.60 \pm 0.05$ | $0.64 \pm 0.02$ | $0.47 \pm 0.05$ | $0.40 \pm 0.03$ | $0.64 \pm 0.05$ | $0.30 \pm 0.02$ |
| GTA | $0.56 \pm 0.05$ | $0.64 \pm 0.02$ | $0.48 \pm 0.05$ | $0.44 \pm 0.03$ | $0.69 \pm 0.05$ | $0.33 \pm 0.02$ |
| DU | $0.55 \pm 0.05$ | $0.67 \pm 0.02$ | $0.48 \pm 0.05$ | $0.40 \pm 0.03$ | $0.65 \pm 0.05$ | $0.32 \pm 0.02$ |
| DU[c] | $\mathbf{0.64} \pm 0.05$ | $0.64 \pm 0.02$ | $0.40 \pm 0.05$ | $0.44 \pm 0.03$ | $0.75 \pm 0.04$ | $0.30 \pm 0.03$ |
| DF | $0.57 \pm 0.05$ | $0.67 \pm 0.02$ | $\mathbf{0.54} \pm 0.05$ | $\mathbf{0.47} \pm 0.03$ | $0.74 \pm 0.04$ | $0.30 \pm 0.02$ |
| DF[c] | $0.62 \pm 0.05$ | $0.65 \pm 0.02$ | $0.44 \pm 0.05$ | $0.45 \pm 0.03$ | $0.73 \pm 0.04$ | $\mathbf{0.34} \pm 0.03$ |
| SU | $0.59 \pm 0.05$ | $0.66 \pm 0.02$ | $0.43 \pm 0.05$ | $0.46 \pm 0.03$ | $\mathbf{0.78} \pm 0.04$ | $0.30 \pm 0.03$ |
| SU[c] | $0.61 \pm 0.05$ | $0.65 \pm 0.02$ | $0.44 \pm 0.05$ | $0.45 \pm 0.03$ | $0.73 \pm 0.04$ | $0.32 \pm 0.02$ |
| SF | $0.59 \pm 0.05$ | $\mathbf{0.67} \pm 0.02$ | $0.48 \pm 0.05$ | $0.46 \pm 0.03$ | $0.70 \pm 0.05$ | $0.30 \pm 0.02$ |
| SF[c] | $0.62 \pm 0.05$ | $0.66 \pm 0.02$ | $0.39 \pm 0.05$ | $0.44 \pm 0.03$ | $0.70 \pm 0.05$ | $0.31 \pm 0.03$ |

Table 7: Llama-3-8B-Instruct Faithfulness for Different In-Context Learning Approaches

| Approach | AQuA | | LogiQA | | TruthfulQA | |
|---|---|---|---|---|---|---|
| | Accuracy | Faithfulness | Accuracy | Faithfulness | Accuracy | Faithfulness |
| ZS-CoT | $0.43 \pm 0.05$ | $0.61 \pm 0.02$ | $0.38 \pm 0.05$ | $0.41 \pm 0.03$ | $0.56 \pm 0.05$ | $0.23 \pm 0.03$ |
| GTA | $0.48 \pm 0.05$ | $0.64 \pm 0.02$ | $\mathbf{0.47} \pm 0.05$ | $0.40 \pm 0.03$ | $0.57 \pm 0.05$ | $\mathbf{0.26} \pm 0.03$ |
| DU | $0.44 \pm 0.05$ | $\mathbf{0.65} \pm 0.02$ | $0.43 \pm 0.05$ | $0.38 \pm 0.03$ | $0.59 \pm 0.05$ | $0.24 \pm 0.03$ |
| DU$^c$ | $\mathbf{0.55} \pm 0.05$ | $0.63 \pm 0.02$ | $\mathbf{0.47} \pm 0.05$ | $\mathbf{0.43} \pm 0.03$ | $\mathbf{0.65} \pm 0.05$ | $0.23 \pm 0.03$ |
| DF | $0.43 \pm 0.05$ | $0.63 \pm 0.02$ | $0.35 \pm 0.05$ | $0.43 \pm 0.03$ | $0.59 \pm 0.05$ | $0.24 \pm 0.03$ |
| DF$^c$ | $0.37 \pm 0.05$ | $0.64 \pm 0.02$ | $0.42 \pm 0.05$ | $0.39 \pm 0.03$ | $0.64 \pm 0.05$ | $0.21 \pm 0.03$ |
| SU | $0.52 \pm 0.05$ | $0.62 \pm 0.02$ | $0.45 \pm 0.05$ | $0.42 \pm 0.03$ | $0.60 \pm 0.05$ | $0.25 \pm 0.03$ |
| SU$^c$ | $0.45 \pm 0.05$ | $0.63 \pm 0.02$ | $0.44 \pm 0.05$ | $0.42 \pm 0.03$ | $0.57 \pm 0.05$ | $0.23 \pm 0.03$ |
| SF | $0.43 \pm 0.05$ | $0.64 \pm 0.02$ | $0.41 \pm 0.05$ | $0.39 \pm 0.03$ | $0.58 \pm 0.05$ | $0.23 \pm 0.03$ |
| SF$^c$ | $0.42 \pm 0.05$ | $0.65 \pm 0.02$ | $\mathbf{0.47} \pm 0.05$ | $0.41 \pm 0.03$ | $0.58 \pm 0.05$ | $0.23 \pm 0.03$ |

Table 8: GPT-3.5-Turbo p-values of Faithfulness for Different Fine-tuning Approaches

| Comparing | AQuA | | LogiQA | | TruthfulQA | |
|---|---|---|---|---|---|---|
| | ZS-CoT | GTA | ZS-CoT | GTA | ZS-CoT | GTA |
| DU | 0.1946 | 0.2247 | 0.0000 | 0.0005 | 0.6353 | 0.1101 |
| DU$^c$ | 0.0718 | 0.0974 | 0.2141 | 0.0597 | 0.3573 | 0.4600 |
| DF | 0.3640 | 0.2610 | 0.0000 | 0.0000 | 0.3607 | 0.4292 |
| DF$^c$ | 0.9523 | 0.7473 | 0.0917 | 0.0201 | 0.2364 | 0.6090 |
| SU | 0.4740 | 0.7740 | 0.0014 | 0.0010 | 0.0473 | 0.9173 |
| SU$^c$ | 0.8063 | 0.9671 | 0.0088 | 0.0028 | 0.2353 | 0.5102 |
| SF | 0.8789 | 0.6707 | 0.0000 | 0.0000 | 0.0579 | 0.9934 |
| SF$^c$ | 0.8324 | 0.6255 | 0.0006 | 0.0001 | 0.1071 | 0.7794 |

Table 9: Llama-3-8B-Instruct p-values for Different Fine-tuning Approaches

| Comparing | AQuA | | LogiQA | | TruthfulQA | |
|---|---|---|---|---|---|---|
| | ZS-CoT | GTA | ZS-CoT | GTA | ZS-CoT | GTA |
| DU | 0.4325 | 0.0062 | 0.0027 | 0.0000 | 0.1835 | 0.1687 |
| DU$^c$ | 0.0845 | 0.0000 | 0.5589 | 0.0000 | 0.7541 | 0.7380 |
| DF | 0.3175 | 0.0103 | 0.1958 | 0.0000 | 0.0130 | 0.0194 |
| DF$^c$ | 0.6068 | 0.0011 | 0.3946 | 0.0000 | 0.0580 | 0.0639 |
| SU | 0.0020 | 0.4670 | 0.1636 | 0.0000 | 0.1311 | 0.1476 |
| SU$^c$ | 0.9323 | 0.0020 | 0.1537 | 0.0000 | 0.7327 | 0.6844 |
| SF | 0.9893 | 0.0003 | 0.2321 | 0.0000 | 0.2940 | 0.2954 |
| SF$^c$ | 0.6049 | 0.0012 | 0.3319 | 0.0000 | 0.1527 | 0.2178 |

Table 10: GPT-4 p-values for Different In-Context Learning Approaches

| Comparing | AQuA | | LogiQA | | TruthfulQA | |
|---|---|---|---|---|---|---|
| | ZS-CoT | GTA | ZS-CoT | GTA | ZS-CoT | GTA |
| DU | 0.3089 | 0.8058 | 0.1395 | 0.7462 | 0.6632 | 0.2648 |
| DU$^c$ | 0.6307 | 0.1890 | 0.4525 | 0.3024 | 0.2392 | 0.5765 |
| DF | 0.5638 | 0.7929 | 0.9936 | 0.1062 | 0.0048 | 0.0489 |
| DF$^c$ | 0.1322 | 0.4820 | 0.3382 | 0.4337 | 0.0250 | 0.2369 |
| SU | 0.6778 | 0.7624 | 0.0509 | 0.8104 | 0.0063 | 0.0572 |
| SU$^c$ | 0.3145 | 0.7599 | 0.3932 | 0.3125 | 0.0297 | 0.2525 |
| SF | 0.2818 | 0.6367 | 0.9038 | 0.0491 | 0.0478 | 0.5067 |
| SF$^c$ | 0.2677 | 0.5417 | 0.0037 | 0.2679 | 0.0011 | 0.0111 |

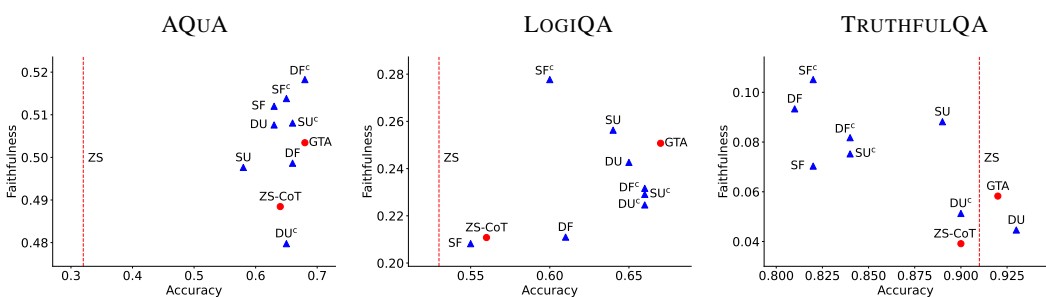

Figure 15: Faithfulness vs Accuracy relationship of CoT reasoning generated by GPT-4 using different baseline (in red) and **ICL** strategies (in blue). Results show that stochastic faithful sampling strategies, on average across three datasets, achieves higher faithfulness in CoT reasoning.

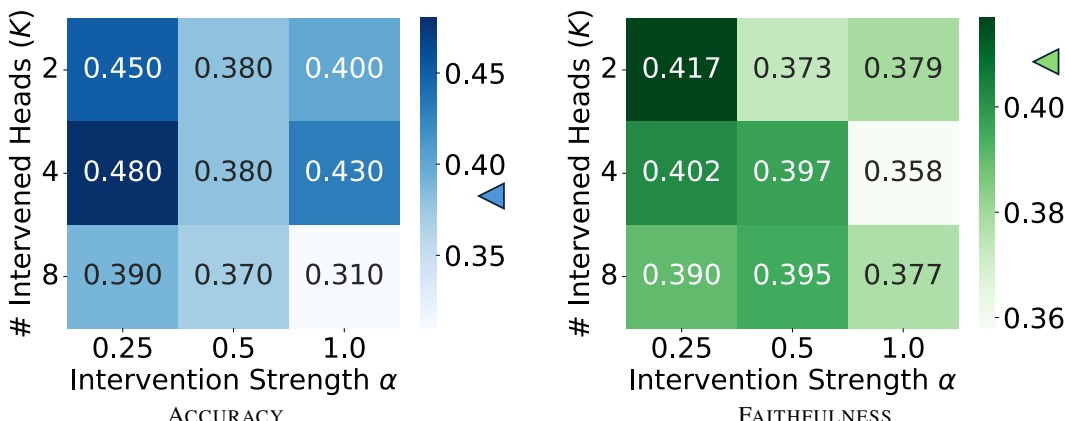

Figure 16: Accuracy and faithfulness of LLM reasoning for different intervention configurations $(\alpha, K)$ for LOGIQA dataset. Activation editing shows different the trade-off between the accuracy and faithfulness performance of LLAMA-3-8B-INSTRUCT and some configuration leads to an increase in accuracy as compared to the zero-shot CoT performance (▲ and ▲ markers) but doesn't improve faithfulness significantly.

Table 11: GPT-3.5-Turbo p-values for Different In-Context Learning Approaches

| Comparing | AQuA | | LogiQA | | TruthfulQA | |
|---|---|---|---|---|---|---|
| | ZS-CoT | GTA | ZS-CoT | GTA | ZS-CoT | GTA |
| DU | 0.2748 | 0.1188 | 0.8037 | 0.1245 | 0.4544 | 0.7770 |
| DU$^c$ | 0.8994 | 0.8539 | 0.1285 | 0.8728 | 0.9518 | 0.3670 |
| DF | 0.1845 | 0.0451 | 0.0144 | 0.3093 | 0.9840 | 0.3429 |
| DF$^c$ | 0.8065 | 0.5908 | 0.0505 | 0.7696 | 0.2248 | 0.7428 |
| SU | 0.4238 | 0.2524 | 0.0186 | 0.5463 | 0.9364 | 0.3364 |
| SU$^c$ | 0.8541 | 0.5486 | 0.0323 | 0.6991 | 0.5434 | 0.6946 |
| SF | 0.0992 | 0.0558 | 0.0093 | 0.3931 | 0.8899 | 0.4127 |
| SF$^c$ | 0.2790 | 0.1526 | 0.1431 | 0.8505 | 0.6492 | 0.6452 |

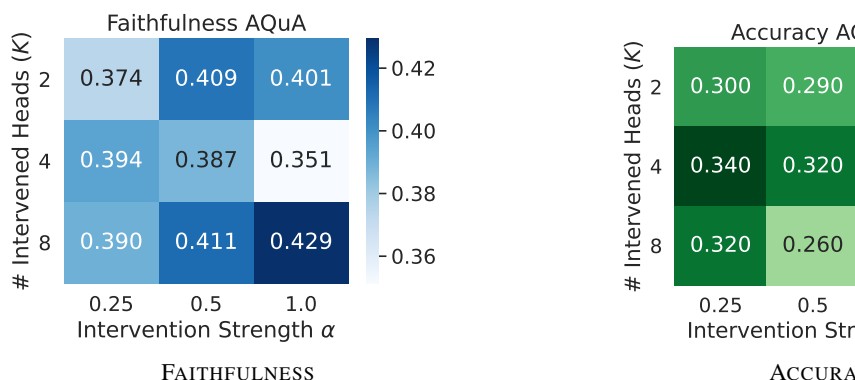

Figure 17: Accuracy and faithfulness of LLM reasoning for different intervention configurations $(\alpha, K)$ on AQUA dataset. Activation editing shows different trade-offs between the accuracy and faithfulness performance of GEMMA-7B-IT. While intervention leads to an increase in faithfulness but has a tradeoff with accuracy.

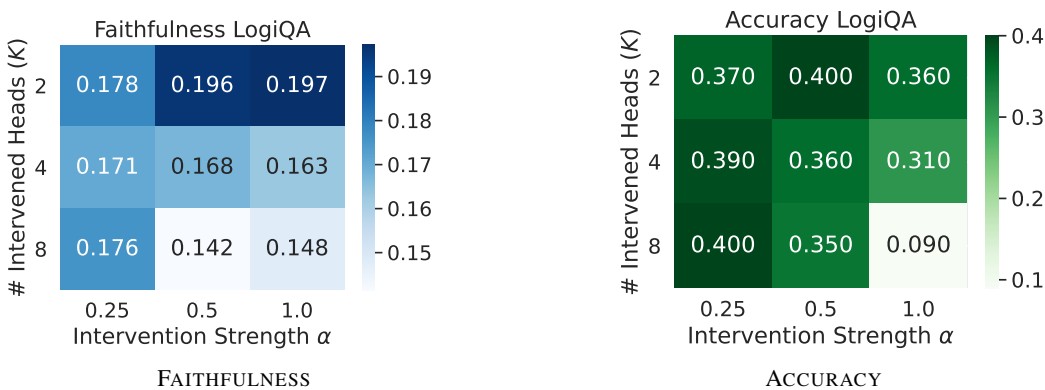

Figure 18: Accuracy and faithfulness of LLM reasoning for different intervention configurations $(\alpha, K)$ on LOGIQA dataset. Activation editing shows different trade-offs between the accuracy and faithfulness performance of GEMMA-7B-IT. While intervention leads to an increase in faithfulness but has a tradeoff with accuracy.

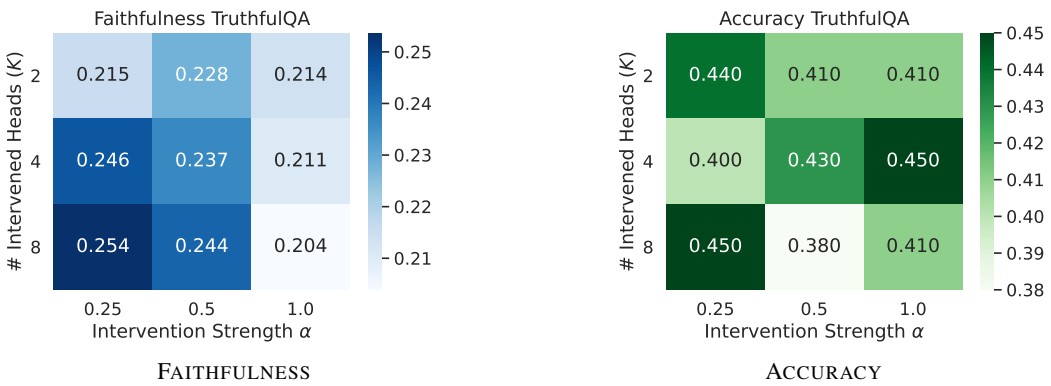

Figure 19: Accuracy and faithfulness of LLM reasoning for different intervention configurations $(\alpha, K)$ on TRUTHFULQA dataset. Activation editing shows different trade-offs between the accuracy and faithfulness performance of GEMMA-7B-IT. While intervention leads to an increase in faithfulness but has a tradeoff with accuracy.

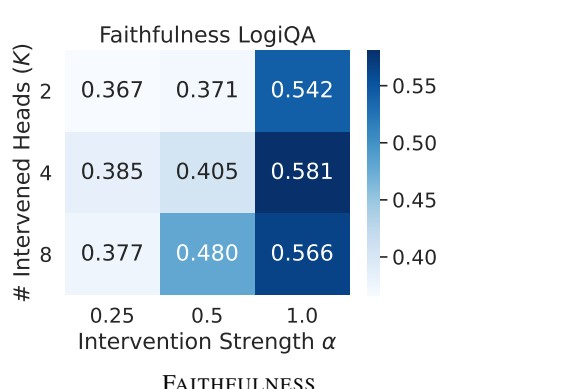
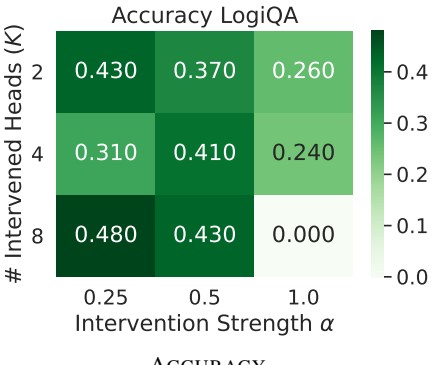

FAITHFULNESS                                    ACCURACY

Figure 20: Accuracy and faithfulness of LLM reasoning for different intervention configurations $(\alpha, K)$ on LOGIQA dataset and MISTRAL-7B-INSTRUCT model. While intervention leads to an increase in faithfulness, there is often a drop in model accuracy.

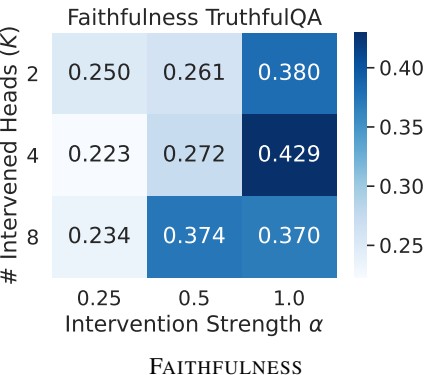
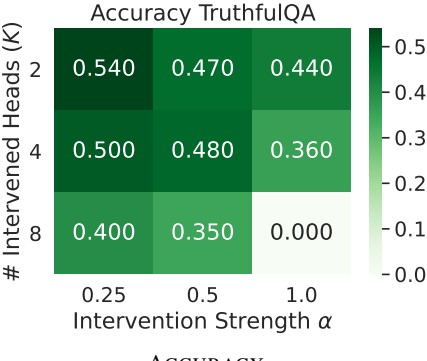

FAITHFULNESS                                    ACCURACY

Figure 21: Accuracy and faithfulness of LLM reasoning for different intervention configurations $(\alpha, K)$ on TRUTHFULQA dataset and MISTRAL-7B-INSTRUCT model. While intervention leads to an increase in faithfulness, it is accompanied by a drop in accuracy.

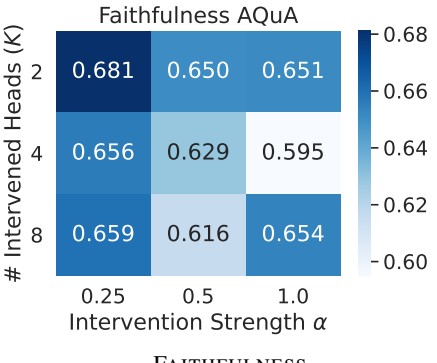
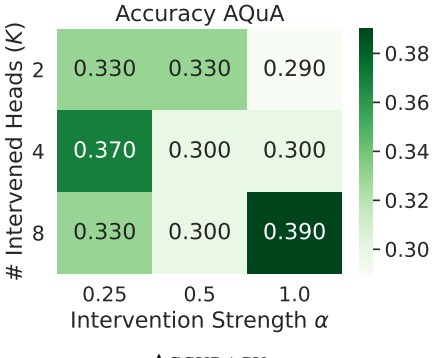

FAITHFULNESS                                    ACCURACY

Figure 22: Accuracy and faithfulness of LLM reasoning for different intervention configurations $(\alpha, K)$ on MISTRAL-7B-INSTRUCT model and AQUA dataset. Contrary to other models, faithfulness doesn't improve upon intervention.

Table 12: Llama-3-8B-Instruct p-values for Different In-Context Learning Approaches

| Comparing | AQuA | | LogiQA | | TruthfulQA | |
|---|---|---|---|---|---|---|
| | ZS-CoT | GTA | ZS-CoT | GTA | ZS-CoT | GTA |
| DU | 0.0488 | 0.5230 | 0.3320 | 0.4825 | 0.8569 | 0.2464 |
| $DU^c$ | 0.2151 | 0.8029 | 0.3341 | 0.2573 | 0.7859 | 0.1824 |
| DF | 0.2610 | 0.7089 | 0.4776 | 0.3101 | 0.6582 | 0.4255 |
| $DF^c$ | 0.2190 | 0.8713 | 0.6078 | 0.8081 | 0.3891 | 0.0104 |
| SU | 0.6302 | 0.2352 | 0.7058 | 0.5657 | 0.5822 | 0.5463 |
| $SU^c$ | 0.3399 | 0.4633 | 0.6561 | 0.5424 | 0.9518 | 0.2365 |
| SF | 0.2268 | 0.8976 | 0.4570 | 0.7079 | 0.9604 | 0.2342 |
| $SF^c$ | 0.1095 | 0.7537 | 0.8032 | 0.6608 | 0.8679 | 0.1552 |

