# OpenReview forum: "On the Hardness of Faithful Chain-of-Thought Reasoning in Large Language Models"
_ICLR.cc/2025/Conference — Submitted to ICLR 2025_

### Official Review · Reviewer_De5a · 2024-10-26

**Soundness:** 3
**Presentation:** 4
**Contribution:** 3
**Rating:** 8
**Confidence:** 2

**Summary:**

The paper examines the difficulty of making large language models produce reasoning that accurately reflects their internal processes. It tests methods like in-context learning, fine-tuning, and activation editing and finds they only marginally improve a model's ability to produce faithful reasoning. The study concludes that current techniques are insufficient to ensure reasoning transparency in language models.

**Strengths:**

1. This paper tackles the issue of enhancing the faithfulness of reasoning in large language models, which is vital for applications requiring high reliability.

2. The study is methodologically sound, with rigorous experiments across different models and datasets, demonstrating the limited effectiveness of current strategies in improving reasoning faithfulness.

3. The findings are impactful, highlighting the need for new methodologies to make LLMs more transparent and trustworthy, which is crucial for their adoption in high-stakes domains.

**Weaknesses:**

1. The study focuses on a limited number of benchmarks. It would benefit from expanding the range of datasets to better understand how these findings generalize across different types of reasoning tasks and domains.

2. The paper could benefit from a more robust theoretical framework that explains why certain strategies might improve faithfulness while others do not.

**Questions:**

Please refer to the Weaknesses

---

> ### Author Response · Authors · 2024-11-25
> **Response to Reviewer De5a**
>
> Thank you for your thoughtful review of our paper. We are delighted that you recognize the importance of the issue of faithfulness, the soundness of our investigation, and the impact of our findings. We would like to address the weaknesses you pointed out and provide clarification on the questions raised.
>
> **W1: Number of benchmarks**
>
> We appreciate the reviewer’s suggestion to expand datasets/domains. This would surely enhance the paper’s original findings, though we do believe that this belongs in future work for two reasons. First, the objective of our study was to demonstrate that conventional strategies are not guaranteed to improve faithfulness, and second, we are interested in reasons why faithfulness is fundamentally hard to achieve. To do so, we emphasized a breadth of systematic investigation across many potential strategies for finetuning or in-context learning example selection, across reasoning vs non-reasoning datasets, in order to find failure cases and explanations as to why. We value the suggestion to explore other domains for additional insights in follow-up work.
>
> **W2: Explanations for success and failure of faithfulness**
>
> This is a very important point, to which we have dedicated our **global response** to. In fact, many explanations we originally had of why faithfulness is hard to achieve, were not prioritized correctly in the main text (see Appendix D). In general, faithfulness is not guaranteed. We thank the reviewer for pointing out the importance of these results for the main text.

---

> > ### Comment · Reviewer_De5a · 2024-11-27
> >
> > Thanks for the response.

---

### Official Review · Reviewer_fxK8 · 2024-10-27

**Soundness:** 2
**Presentation:** 3
**Contribution:** 1
**Rating:** 3
**Confidence:** 3

**Summary:**

This paper investigates the challenge of generating faithful Chain-of-Thought reasoning in large language models, specifically focusing on approaches like in-context learning, fine-tuning, and activation editing. While the authors highlight the importance of faithfulness in CoT reasoning for trustworthiness in high-stakes domains like healthcare, their empirical results suggest that none of these methods yield significant improvements in CoT faithfulness.

**Strengths:**

(1) The topic of faithful reasoning of LLMs is interesting and sounds reasonable to investigate.

(2) By demonstrating the limited success of conventional strategies, the paper highlights the intrinsic difficulty of faithful reasoning in LLMs, which provides a strong basis for future exploration. ​

(3) The presentation is generally clear and easy to follow.

**Weaknesses:**

(1) The paper evaluates standard techniques like in-context learning, fine-tuning, and activation editing to improve CoT faithfulness, but these methods have already been extensively studied in other contexts such as improving accuracy, bias reduction, and factual consistency. The paper does not present any substantial technical adaptations or theoretical contributions to these methods specifically for faithful CoT reasoning. For example, while activation editing is discussed, it largely follows the framework of existing works like Li et al. (2024) without offering any new insights. The novelty of merely applying them to faithful CoT seems limited, and the contribution does not significantly advance the field beyond the status quo.

(2) The "early answering" metric used to evaluate faithfulness is based on whether truncating CoT reasoning affects the model's final output. However, the reason for taking it as the best way to measure faithfulness remains unclear, particularly given the complexity of CoT explanations. The measure seems too simplistic, as it fails to capture nuances in reasoning that may be faithful but not necessarily immediately reflected in the final answer. This could raise a misalignment between the metric and the goal of the research, which is to assess whether CoT explanations reflect the internal logic of the LLM.

(3) Although the paper acknowledges that none of the explored methods significantly improve CoT faithfulness, it does not provide a deep analysis of why these methods fail. For example, the results show only marginal gains in faithfulness, but the paper does not dive into what specifically causes this limitation—whether it is the inherent architecture of LLMs, the quality of training data, or other factors.

(4) While the paper claims to "lay the groundwork" for future research in trustworthy CoT reasoning, it does not propose concrete next steps or actionable insights based on the experimental findings. The conclusion merely restates that current methods are insufficient without suggesting innovative ideas or frameworks that could be explored in the future. This lack of direction limits the potential impact of the paper in advancing the field.

**Questions:**

As the authors claim that "our work underscores the inherent difficulty in eliciting faithful CoT reasoning from LLMs, suggesting that the current array of approaches may not be sufficient to address this challenge", I wonder what could be revealed from the evaluation about the fundamental cause of the limitation for current LLM paradigms? Further, what could be the potential way to address them?

---

> ### Author Response · Authors · 2024-11-25
> **Response to Reviewer fxK8 (1/2)**
>
> Thank you for your thoughtful review of our paper. We are glad to hear that you believe our paper provides a strong basis for future exploration. We would like to address the weakness you pointed out and provide clarification.
>
> **W1/W4: Application of existing techniques and concrete next steps**
>
> We appreciate the reviewer’s viewpoint that we are utilizing existing techniques. The main purpose of our investigation is indeed to assess the abilities of *existing* intervention strategies, specifically on faithfulness. Ensuring the faithfulness of LLM-generated CoT reasoning is crucial for decision-makers, such as doctors, who rely on them to determine if, when, and how much to trust the recommendations made by these LLMs. It is therefore important to understand when and why these methods fail. The key contribution of our work is to explore if popular approaches that found success in modifying LLM outputs to improve properties like accuracy [1] and truthfulness [2] can improve the faithfulness of CoT reasoning generated by LLMs. Previous seminal works [3-6] have also explored whether common strategies are effective at solving certain problems across domains. **The novelty and significance of these kinds of papers do not stem from a new approach or method** but rather the significant insight that none of the popular approaches work for this problem, and we need a new set of approaches.
>
> We are happy to release our coding framework (supplemental material) upon publication, which includes extensive pipelines that lay the groundwork for assessing faithfulness, using both OpenAI APIs and Hugging Face GPU implementations. Please note that this is a rapidly evolving field with many metrics / understandings of faithfulness and our work provides a strong datapoint upon which to build, and performing evaluations so extensively is not cheap. As a result, we uncover in-depth insights concerning the limitations of ICL (word mimicking, incentivization of late change in reasoning), and the fundamental uniqueness of trying to finetune for faithfulness (please see W3 below and our **global response**).
>
> **W2: On the particular faithfulness metric used**
>
> Measuring faithfulness of reasoning without having access to a black box is not straightforward, and hence, several works propose tests to evaluate faithfulness. Note that each test only evaluates an explanation of a particular property. We use the early answering test proposed by Lanham et. al. (2023) to measure faithfulness. The premise is that if reasoning is not post-hoc, there are fewer ways for it to be unfaithful than there are for reasoning which is post-hoc. While there are other possible faithfulness measures, they have their limitations as shown below (Table 1, Appendix).
>
> | Strategy | Description | Limitations |
> | ----------- | --------------- | --------------- |
> | Counterfactuals | If features referenced by an explanation are removed, then the model's prediction should change. | More relevant for feature importance explanations than CoT. |
> | Adding Mistakes | If inserting a mistake into the CoT changes the model's final answer, then the model is likely not ignoring the CoT. | Dependent on external factors of generating mistakes, which influences faithfulness values. Difficult to ablate across mistakes. |
> | Paraphrasing	 | If information encoded in phrasing choices of the reasoning are responsible for the change in the answer, rather than the content of the CoT itself, then the CoT is unfaithful. | Dependent on external factors to paraphrase steps, which influences faithfulness values. Difficult to ablate across paraphrases. |
>
> Unlike the 'Adding Mistakes', and 'Paraphrasing' strategies, the 'Early Answering' strategy uses the generated CoT only from the model to measure faithfulness, thereby avoiding reliance on an external model/mechanism to evaluate faithfulness. However, we also looked at the 'Adding Mistakes' and 'Paraphrasing' strategies to measure faithfulness and found that both these measures are highly correlated with faithfulness from the 'Early Answering' strategy, shown in Fig 11 (Appendix). Our observation is also consistent with the finding reported in Lanham et. al. (2023).
>
> Thank you for your comments. We will move Table 1 (above) and Fig 11 from the appendix to the main text, to prioritize the reviewer’s concerns (they remain in the appendix for now for identification purposes).

---

> ### Author Response · Authors · 2024-11-25
> **Reviewer fxK8 (2/2)**
>
> **W3/Q1: Why is faithfulness fundamentally challenging?**
>
> This is a great point, and one that we did not highlight enough in the main text, though we have thought about it extensively. We kindly invite the reviewer to read our global response. The goal of faithful Chain of Thought (CoT) reasoning involves fundamentally altering the model to generate reasoning that is more consistent with its internal decision-making processes. This represents an intrinsic change in the model's behavior, rather than simply learning a new task. When trying to finetune based on faithful CoT examples, there is no guarantee that such examples remain faithful once model parameters are adjusted. In other words, supervised learning for faithfulness poses non-stationary/moving targets. We provide a carefully designed experiment to demonstrate empirical evidence of this. As it turns out, we made findings on the fundamental challenges of finetuning for faithfulness and the limitations of ICL, in Appendices D3 and D4, but failed to prioritize their relevance correctly for the main text. We therefore thank the reviewer sincerely for bringing this to our attention.
>
> **Experiment:** we measure the faithfulness of the finetuned model’s CoT explanations with respect to both itself and the original model. This does not make sense in other domains with static datasets, but for faithfulness is crucially insightful, where there do not exist ground truth faithful (question, explanation) pairs. What our demonstrations strongly suggest is that pairs that are faithful w.r.t. the original model may no longer be guaranteed to be faithful w.r.t. a finetuned model, supporting our claim that **faithfulness is fundamentally difficult to target using supervised learning.**
>
> **Closing Remarks**
>
> Thank you again for your fruitful feedback! We would like to invite you to further discussion, in case your concerns are still not addressed. It would be great if you can let us know if you have any additional concerns, and we will be happy to respond. Should the insights satisfy your queries on why faithfulness is difficult to learn, we would strongly appreciate your vote of acceptance.
>
> [1] Wei et al. Emergent abilities of large language models. TMLR, 2022.
>
> [2] Li et al. Inference-Time Intervention: Eliciting Truthful Answers from a Language Model. NeurIPS, 2023.
>
> [3] Goodfellow et al. Explaining and Harnessing Adversarial Examples. ICLR, 2015.
>
> [4] Bolukbasi et al. Man is to Computer Programmer as Woman is to Homemaker? Debiasing Word Embeddings. NeurIPS, 2016.
>
> [5] Adebayo et al. Sanity Checks for Saliency Maps. NeurIPS, 2018.
>
> [6] Jain et al. Attention is Not Explanation. ACL, 2019.

---

> > ### Comment · Reviewer_fxK8 · 2024-11-27
> >
> > Thanks for the authors' global responses and specific explanations towards my initial questions. After reading all the responses as well as feedbacks from other reviewers, I still have some concerns about this work.
> >
> > While the paper explains why certain alternative metrics (e.g., counterfactuals or paraphrasing) were not chosen, it remains unclear why “early answering” is a reasonable metric for measuring faithfulness, particularly for black-box LLMs. Specifically because: (1) Faithfulness, as noted in the authors’ response, lacks a clear formal definition due to the complexity of internal reasoning in LLMs (“What constitutes internal reasoning for LLMs is a vastly complex topic, and as such can’t be reduced to a simple formal definition”). Without a clear foundation, how could the authors ensure that "early answering" meaningfully represents faithfulness? (2) For the metric itself, the idea that truncating reasoning steps alters a model's outputs is intriguing but does not convincingly establish that this alteration correlates with internal faithfulness rather than being an incidental behaviour.
> >
> > As I understand it, this paper mainly reports the results of existing techniques including in-context learning, fine-tuning and activation editing to control the “early answering”. Such a “finding” is not novel or solid enough to sorely make the paper get accepted. Not only as some insights could already be seen from the previous work by Lanham et. al. (2023), but also the less-satisfying results of CoT or fine-tuning are more or less expectable, as those techniques were not initially designed for "not early answering", but for improving the end-to-end performance. Beyond that, the paper did not provide either clear approaches to improve the so-called “faithfulness” or fundamental analyses about what caused this issue of unfaithfulness.
> >
> > In this sense, I would agree with the point raised by reviewer qPR7 that the paper did not go deep enough, but more in the aspects that (1) what essentially constitutes the LLMs’ faithfulness and how should we measure it; and (2) what is the best (provable) practice to improve faithfulness, which should not be limited by existing techniques like CoT or fine-tuning.
> >
> > Given all these concerns, I would maintain my current score unless further evidence is provided that are more reasonable or convincing. I would encourage the authors to expand their analysis and propose more innovative approaches in future iterations of this work, and thanks again for the efforts.

---

### Official Review · Reviewer_qPR7 · 2024-11-03

**Soundness:** 3
**Presentation:** 2
**Contribution:** 2
**Rating:** 3
**Confidence:** 3

**Summary:**

The paper did an empirical study on whether chain of thought reasoning can be made to accurately reflect the underlying reasoning done by the LLM (i.e. whether it can be made faithful) by in-context learning, fine-tuning, or activation editing. The faithfulness measurement tries to measure whether stopping the chain of thought early would results in different outcomes compared to using the full chain to answer the question; if it does not, it is an indication that the LLM already knows the answer before generating the chain and is doing post-hoc explanation of its reasoning in the chain rather than computing the answer within the chain. The study found that in-context learning, fine-tuning, and activation editing are all not successful in substantially improving faithfulness.

**Strengths:**

The paper provided experimental results indicating that in-context learning, fine-tuning, and activation editing did not result in substantial improvement in faithfulness of chain of thought reasoning. This suggests that other techniques may be required if this type of faithfulness is required.

**Weaknesses:**

The paper provides negative results -- this is fine. However, to make a strong paper, insights that are supported by evidence on why the results are negative would be helpful.

**Questions:**

Insights on why faithfulness is difficult to learn, either in the form of mathematical theorems, or carefully designed experiments would be helpful.

---

> ### Author Response · Authors · 2024-11-25
> **Response to Reviewer qPR7**
>
> Thank you for your thoughtful review of our paper, and for recognizing our main contribution that other intervention strategies may be required to achieve faithfulness, based on our investigation of existing techniques. We would like to address the weakness you pointed out and provide clarification.
>
> **W1/Q1: Insights on why faithfulness is difficult to learn, either in the form of mathematical theorems, or carefully designed experiments would be helpful**
>
> This is an important point and one that we did not highlight enough in the main text, though we have thought extensively about it. The goal of faithful Chain of Thought (CoT) reasoning involves fundamentally altering the model to generate reasoning that is more consistent with its internal decision-making processes. This represents an intrinsic change in the model's behavior, rather than simply learning a new task. This poses significant challenges when trying to learn faithful CoT reasoning, since there is no guarantee that any faithful examples used will remain faithful once model parameters are adjusted. In other words, learning faithfulness poses non-stationary/moving targets.
>
> **New experiments**
>
> We are happy to provide a carefully designed experiment to demonstrate empirical evidence of this. Please refer to our **global response** for full details. We measure the faithfulness of the finetuned model’s CoT explanations with respect to both itself and the original model. This does not make sense in other domains with static datasets, but for faithfulness is crucially insightful, where there do not exist ground truth faithful (question, explanation) pairs. Our findings demonstrate that once finetuning occurs, model internals shift, and what constitutes faithfulness changes. This is analogous to chasing a dynamic/moving target and poses fundamental challenges to standard static finetuning.
>
> As it turns out, we made findings on the fundamental challenges of finetuning for faithfulness and the limitations of ICL, in Appendices D3 and D4, but failed to prioritize their relevance correctly for the main text. Therefore, we thank the reviewer sincerely for bringing this to our attention. We have dedicated a section in the main text to these findings.
>
> **Closing Remark**
>
> It would be great if you can let us know if you have any additional concerns, and we will be happy to respond. Should the insights satisfy your queries on why faithfulness is difficult to learn, we would strongly appreciate a vote of acceptance for our paper.

---

> > ### Comment · Reviewer_qPR7 · 2024-11-27
> >
> > Thank you for your response and additional explanation on why fine-tuning and in-context learning did not help faithfulness in your work. As I understand it, the loss function used for fine-tuning in the work aims to improve the accuracy in explanation and the final answer, and not for improving faithfulness. Given that, it is not surprising that it did not improve faithfulness. I would maintain my view that the paper did not go deep enough in providing understanding on whether faithfulness is difficult to learn, even if you were to design methods, e.g. loss functions, for learning it.

---

> > > ### Author Response · Authors · 2024-11-27
> > >
> > > Thank you for your response. We greatly appreciate your engagement in this discussion.
> > >
> > > **To clarify:** we do not finetune to improve the *accuracy* of explanations, rather, we finetune the model to output explanations that are *faithful* w.r.t. the model. Faithfulness is the primary driver for the examples, to steer the model towards more faithful CoT reasoning.
> > >
> > > For example, the **Stochastic Uniform (SU)** selection strategy (Section 3.2, line 245 ) works as follows:
> > > 1. We sample 10 CoT responses (at temperature 0.3) for every question in the training set.
> > > 2. For each question, we pick the most faithful CoT response.
> > > 3. We then finetune the model on these faithful (question, explanation) pairs.
> > >
> > > As per the first table in our global response, we can observe that **faithfulness w.r.t. the original model does improve** from 0.6 to 0.65 on the AQuA dataset using this strategy. However, faithfulness w.r.t. the resulting finetuned model decreases to 0.584. We demonstrate that the fundamental limitation of finetuning for faithfulness is that, unlike accuracy, it is a property intrinsic to the model, and thus *any* loss function operating on the model's outputs alone can bear no guarantee on the new model.
> > >
> > > For this reason, our work's scope is to demonstrate that existing data-driven (ICL and finetuning) and activation editing (ITI) frameworks cannot steer LLMs to automatically generate faithful explanations. Our research provides a useful datapoint for practitioners that faithfulness is not a property that can be achieved using a completely data-driven approach like supervised finetuning.
> > >
> > > Furthermore, we want to point out that this exploration has not been previously performed. It is unclear at first whether finetuning a model on faithful explanations would lead to reliable improvements in faithfulness, and our findings demonstrate that model faithfulness is particularly more nuanced.
> > >
> > > We hope that this helps to clarify our experimentation! Please let us know if you have questions about our response. Thank you for your time.

---

### Official Review · Reviewer_gNfW · 2024-11-04

**Soundness:** 3
**Presentation:** 3
**Contribution:** 2
**Rating:** 5
**Confidence:** 4

**Summary:**

The paper systematically examines the Chain-of-Thought (CoT) behavior in large language models (LLMs) through experiments involving in-context learning, fine-tuning, and activation editing. The results indicate that activation editing had limited success, while in-context learning and fine-tuning led to only slight, non-generalizable improvements. The authors argue that the training process of these models does not prioritize faithfulness, which contributes to the generation of more self-aware content.

**Strengths:**

1. **Detailed Experiment** The paper conducted thorough experiments across in-context learning, fine-tuning, and activation editing.

2. **Insights from the Experiment** The empirical experiments provided meaningful insights, which might inform a better LLM alignment methodology for achieving faithful intermediate states in the future.

**Weaknesses:**

**Novelty in methodology** is the weakness of this position paper. As author(s) stated, changes are made for referenced methods/procedures, some theoretical/numerical supports can better validate the proposal. Please let me know if the following points are biased.

Here are some directions to consider:
1. To measure faithfulness, the Area Over the Curve (AOC) metric from [1] is adopted while the paper proposed to use probability scores for each instance instead of on the dataset level. However, section 2.3.1 of [1] also stated "AOC values are calculated as a weighted sum", thus [1] should also work on the instance level. I suggest editing line 166 to prevent confusion if this is the case.
2. For activation editing, this work selected top-K heads based on faithful probing results instead of top-K truth-relatedness heads in [2], they sound serving similar purposes to me. Can we compare these methods or see if they are transferable?

Reference
[1] Lanham, T., Chen, A., Radhakrishnan, A., Steiner, B., Denison, C., Hernandez, D., ... & Perez, E. (2023). Measuring faithfulness in chain-of-thought reasoning. arXiv preprint arXiv:2307.13702.
[2] Li, K., Patel, O., Viégas, F., Pfister, H., & Wattenberg, M. (2024). Inference-time intervention: Eliciting truthful answers from a language model. Advances in Neural Information Processing Systems, 36.

**Questions:**

1. What is the sample size of the benchmark? Correct me if I am wrong but lines 339 - 348 describe original datasets' statistics instead.
2. When selecting N ICL demonstrations, are we considering questions' similarities or just using faithfulness as the single index?

Minor:
1. Figures' notation requires browsing around.
2. Please avoid directly using acronyms, a full expression would be more reader-friendly. e.g. out of distribution for OoD in line 303
3. Please check typos in the manuscript, such as:
a. line 312, Figure 4?
b. line 354 asking the question *without* invoking?

---

> ### Author Response · Authors · 2024-11-25
> **Response to Reviewer gNfW (1/2)**
>
> Thank you for your thoughtful review of our paper, and for recognizing the thoroughness of our experimental evaluations. We are glad that you found the experiments to be insightful (**please see our global response** for further insights we have attained for finetuning and in-context learning). We would like to address the weaknesses you pointed out and provide clarification on the questions raised.
>
> **W1: Novelty of our contribution**
>
> We appreciate the reviewer’s perspective on the novelty of the methodology. The key contribution of our work is to explore if popular approaches that found success in modifying LLM outputs to improve properties like accuracy [1] and truthfulness [2] can improve the faithfulness of CoT reasoning generated by LLMs. Previous seminal works [3-6] have also explored whether common strategies are effective at solving certain problems across domains. **The novelty and significance of these kinds of papers do not stem from a new approach or method** but rather the significant insight that none of the popular approaches work for this problem, and we need a new set of approaches that can address this problem. We illustrate that in-context learning, finetuning and activation editing offer limited success, and we also provide evidence as to why these methods are (fundamentally) limited in the context of faithfulness.
>
> **W2: AOC values are *not* calculated as a weighted sum as in Lanham et. al. (2023)**
>
> We would like to correct a potential misunderstanding here. In Lanham et. al. (2023), the “weighted sum” for AOC corresponds to a) grouping all CoTs by length, b) averaging across all instances in each group to create one profile per group, c) computing AOC for each group’s profile and d) computing a “weighted sum” across all groups. In other words, as in Lanham et. al. (2023): the AOC for each CoT length is weighted by the fraction of CoT samples having that length. This average the CoT graphs for each group length and then compute AOC for each, whereas our scores represent average AOC across all CoTs (instance-level). We hope this provides clarity! We are actively updating the text to clarify this.
>
> **W3: Can we compare Top-K faithfulness heads and Top-K truthfulness heads?**
>
> The work on truthfulness [2] is an experiment on a factual question answering dataset i.e.; TruthfulQA which contains questions like “What is the capital of France ?” for which there is a single correct answer in the options provided. Such tasks don’t benefit significantly from step by step reasoning. Top-K truthful heads in [2] correspond to the top attention heads whose representations are best predictors of an answer grounded in world knowledge. On the contrary, faithfulness is grounded in the model's inner workings. The exact same reasoning chain generated by two different models can have different values of faithfulness. Hence, top-K truthful heads and top-K faithful heads are different. Upon empirical evaluation, we observe the same. Following table shows the overlap between top-$K$ truthful and faithful heads for Llama-3-8B-Instruct model for varying $K$.
>
> | $K$ | Intersection of top-$K$ Faithful & top-$K$ Truthful heads |
> |:----------------:|:-------------------------------------------:|
> | 4              | 1                                          |
> | 8              | 1                                          |
> | 16             | 2                                          |
> | 32             | 4                                          |
> | 64             | 10                                         |
> | 128            | 25                                         |
> | 256            | 107                                        |
> | 512            | 279                                        |

---

> ### Author Response · Authors · 2024-11-25
> **Response to Reviewer gNfW (2/2)**
>
> **Q1: What is the sample size of the benchmark?**
>
> The sample size of each benchmark was 400 training examples from which to select responses (with 10 responses sampled per question in the case of 0.3 temperature sampling), and 100 test questions per dataset on which to evaluate faithfulness. This is motivated by a) high API costs and b) valuing breadth of evaluation across the several strategies that we introduce, rather than depth in test sample size (our evaluations show that mean test set faithfulness converged, on average, after around 60 to 80 of the 100 samples).
>
> **Q2: When selecting N ICL demonstrations, are we considering questions' similarities or just using faithfulness as the single index?**
>
> Faithfulness was considered as the index for selecting examples for ICL/FT, rather than similarity, since we effectively create a dataset of faithful question-explanation pairs from which the model can learn. Interestingly, this results in more faithfulness to the original model where the examples are initially drawn from (global response). There is no guarantee that finetuning on a model’s faithful or similarly related examples would result in improved faithfulness for the finetuned model. We appreciate the suggestion on similarity, though we believe it belongs squarely in future work.
>
> **Q3-5: Text corrections**
>
> We have updated the figure captions to reduce the requirement to browse around, thank you for the suggestion! We have also updated acronyms as suggested and fixed the mentioned typos, and will upload the new text soon once other edits are complete. We wholeheartedly appreciate the reviewer’s careful examination of our text.
>
> Thank you once again for your insightful suggestions and comments. We are actively incorporating the above explanations to enhance the quality of our paper. We believe that we have addressed all the concerns. If there is any aspect that you feel has not been fully resolved, we would be happy to provide further information. If you are satisfied with our response, we would truly appreciate your vote of acceptance for our paper.
>
> [1] Wei et al. Emergent abilities of large language models. TMLR, 2022.
>
> [2] Li et al. Inference-Time Intervention: Eliciting Truthful Answers from a Language Model. NeurIPS, 2023.
>
> [3] Goodfellow et al. Explaining and Harnessing Adversarial Examples. ICLR, 2015.
>
> [4] Bolukbasi et al. Man is to Computer Programmer as Woman is to Homemaker? Debiasing Word Embeddings. NeurIPS, 2016.
>
> [5] Adebayo et al. Sanity Checks for Saliency Maps. NeurIPS, 2018.
>
> [6] Jain et al. Attention is Not Explanation. ACL, 2019.

---

> > ### Comment · Reviewer_gNfW · 2024-11-25
> >
> > Thank you authors for the point-to-point and global comment. I have no outstanding concerns left and have raised my points accordingly.

---

### Official Review · Reviewer_bkgd · 2024-11-04

**Soundness:** 2
**Presentation:** 2
**Contribution:** 1
**Rating:** 5
**Confidence:** 4

**Summary:**

Recent advances of foundation models, in particular Large Language Models (LLMs) have demonstrated impressive performances in many natural language processing tasks. Nevertheless the capabilities of LLMs at reasoning tasks are still limited and raises significant debate (see [1]). A line of recent works proposed prompt-based techniques to improve LLM capability including, but not limited to, reasoning.  Notably, the most popular techniques are: *chain-of-thought* (CoT) by adding the phrase  'think/solve step by step' at the end of the prompt, and *in-context learning* by including illustrative examples in the prompt to inspire or assist the LLM about the specific context of the query to solve; another line focuses on fine-tuning the LLM on formal reasoning benchmarks data, mathematical problems (Algebra, Geometry, calculus and so on).

This work combines the three aforementioned techniques to improve LLMs in producing what is referred to as *faithful* CoT reasoning and rational explanations to the delivered output. Moreover, it define a metric to assess the concept of faithful CoT reasoning.



[1] Iman Mirzadeh, Keivan Alizadeh, Hooman Shahrokhi, Oncel Tuzel, Samy Bengio, Mehrdad Farajtabar: GSM-Symbolic: Understanding the Limitations of Mathematical Reasoning in Large Language Models. EMNLP 2024

**Strengths:**

The paper targets an important and timely challenge in LLM. While massive effort is dedicated towards enhancing LLM's capability to reason or even demonstrating that it can reason, it still represents a major bottleneck and prevents using LLM to create AGI.

Overall, the paper reads well and is well organised. The overall contribution is more technical and focuses on empirical studies of various combined techniques implemented, resulting in comprehensive experimental evaluations.

**Weaknesses:**

- The  paper constitutes incremental research work. The proposed solution is simply combining applied techniques in LLMs, which render the contribution incremental and straightforward. Technically, the contribution lacks in rigor, and many of the applied strategies are not formally justified.

- The aforementioned techniques have shown several limitations, in past works, and more importantly in many cases techniques like activation patching are deteriorating the LLMs accuracy.

- Several notions and techniques that this work builds upon, are not formally defined or described earlier in the paper, making it less accessible to a broader audience.

**Questions:**

- What is the formal definition of faithful (CoT) reasoning in LLMs? Unless, I am missing something this was stated to be formally defined in line 93, but I fail to find this definition later in the manuscript.

---

> ### Author Response · Authors · 2024-11-25
> **Response to Reviewer bkgd**
>
> Thank you for your thoughtful review of our paper and for recognizing the scope of our comprehensive experimental evaluations, aimed at uncovering the reasons behind limitations in existing intervention strategies for faithfulness. We would like to address the weaknesses you pointed out and provide clarification on the questions raised.
>
> **Q1: What is the formal definition of faithful (CoT) reasoning in LLMs?**
>
> Faithfulness represents how well a model’s expressed reasoning matches its internal reasoning. What constitutes internal reasoning for LLMs is a vastly complex topic, and as such can’t be reduced to a simple formal definition. It varies greatly across different faithfulness metrics (see lines 141-148). We will emphasize this aspect in our text to make it clearer to readers. Thank you pointing this out.
>
> **W1:  The proposed solution is simply combining applied techniques in LLMs, which render the contribution incremental and straightforward**
>
> We appreciate the reviewer’s viewpoint that we are utilizing existing techniques. The main purpose of our investigation is indeed to assess the abilities of *existing* intervention strategies, specifically on faithfulness. Ensuring the faithfulness of LLM-generated CoT reasoning is crucial for decision-makers, such as doctors, who rely on them to determine if, when, and how much to trust the recommendations made by these LLMs. It is therefore important to understand when and why these methods fail. Previous seminal works [1-4] have also explored whether common strategies are effective at solving certain problems across domains. **The novelty and significance of these kinds of papers do not stem from a new approach or method** but rather the significant insight that none of the popular approaches work for this problem, and we need a new set of approaches that can address this problem.
>
> The novelty in our contribution is the following. We illustrate that these methods achieve limited success, and we also provide significant evidence as to why these methods are (fundamentally) limited in the context of faithfulness. **We kindly refer the reviewer to our global response for details**. We in turn systematically evaluate across several novel example selection strategies for ICL/FT, finding that none are able to consistently improve faithfulness across domains. While conventional finetuning appears promising, faithfulness is a unique type of property that assesses a model’s internals and cannot be learnt as a supervised task.
>
> **W2: The aforementioned techniques have shown several limitations, in past works**
>
> Thank you for the feedback. We would like to clarify that fine-tuning and in-context learning are the widely used techniques to adapt pre-trained LLMs to specialized downstream tasks [5-10]. While we agree that there are existing limitations of the aforementioned techniques, it would be great if the reviewer could clarify which limitations would pose known challenges to faithful CoT reasoning.
>
> **W3: Several notions and techniques that this work builds upon, are not formally defined or described earlier in the paper, making it less accessible to a broader audience**
>
> We appreciate the reviewer’s feedback regarding the formal definition of CoT. We are actively resolving this in the updated text. If the reviewer could kindly provide specific examples of ambiguity we would be more than happy to address them and use them to strengthen our paper’s readability. Thank you for helping us improve our writing.
>
> We are actively incorporating the above explanations to enhance the quality of our paper. We believe that we have addressed all the concerns. If there is any aspect that you feel has not been fully resolved, we would be happy to provide further information. If you are satisfied with our response, we would truly appreciate your consideration in raising your evaluation score.
>
> **References**
>
> [1] Goodfellow et al. Explaining and Harnessing Adversarial Examples. ICLR, 2015.
>
> [2] Bolukbasi et al. Man is to Computer Programmer as Woman is to Homemaker? Debiasing Word Embeddings. NeurIPS, 2016.
>
> [3] Adebayo et al. Sanity Checks for Saliency Maps. NeurIPS, 2018.
>
> [4] Jain et al. Attention is Not Explanation. ACL, 2019.
>
> [5] Dettmers et al. Qlora: Efficient finetuning of quantized llms. 2023.
>
> [6] Hu et al. LoRA: Low-rank adaptation of large language models. In ICLR, 2022.
>
> [7] Jeong et al. Domain-specialized llm: Financial fine-tuning and utilization method using mistral 7b. In Journal of Intelligence and Information Systems, 2024.
>
> [8] Kumar et al. Fine-tuning, quantization, and llms: Navigating unintended outcomes. arXiv, 2024.
>
> [9] Rafailov et al. Direct preference optimization: Your language model is secretly a reward model. In NeurIPS, 2023.
>
> [10] Singh et al. Whispered tuning: Data privacy preservation in fine-tuning llms through differential privacy. Journal of Software Engineering and Applications, 2024.

---

### Official Review · Reviewer_hqPm · 2024-11-04

**Soundness:** 3
**Presentation:** 3
**Contribution:** 3
**Rating:** 6
**Confidence:** 4

**Summary:**

In recent years, there have been concerted effort in making language models more faithful and robust with methods such as finetuning, in-context learning and activation editing. This work investigates whether these 3 methods can make CoT reasoning more faithful. Their findings suggest that all of them achieve very limited performance improvements, with activation editing achieving only minimal improvements. Finetuning and in-context learning can be slightly more effective, though they seem to fail to generalize across tasks.

**Strengths:**

Overall, I thought the paper was strong and has potential for broad impact, because it connects so many concepts that are disparately considered. There has been a significant gap in evaluating *for* interventions, and this work systematically investigates the common and practical techniques for interventions.
- S1. Comprehensive evaluation of intervention methods for a widely used technique, CoT reasoning. Since this is how many researchers as well as practitioners interact with LLMs, this work is widely applicable and can have broad impact in considerations for AI safety.
- S2. I thought the introduction was particularly well motivated and the paper was generally well written.
- S3. Finetuning strategies were tested with multiple sampling strategy of their design. Adding faithfulness metric to the finetuning dataset creation was a particularly convincing experimental strategy.
- S4. Also introduces novel strategy for activation editing based on aligning on faithfulness vectors
- S5. The paper includes salient results, with most of these methods getting partial success. ICL or activation editing seem to get either accuracy or faithfulness performance enhancements, but rarely both. It seems that more finetuning on faithful datasets can improve both more so than ICL and activation editing

**Weaknesses:**

- W1. It seems that activation editing was only experimented with LLaMA-3B. I wonder if this could have been an issue with this particular model, particularly because activation editing could have vastly different results depending on the architecture. For that reason, I think this result could be made more robust by adding other models for comparison such as Gemma or OLMo.

- W2. "Fine-tuning using most faithful explanations achieve better accuracy-faithfulness trade-offs." This seems like an expected result, but I wonder if this holds true across domain. If there could have been a more comprehensive strategy such as sampling by length for comparison, I wonder if there were any observable differences across domain.

- W3. There's slew of methods proposed by lanham et al, but I think this paper only discusses faithfulness with respect to early answering strategy. Faithfulness metric could result in different behavior based on the metric definitions: early answering vs. adding mistakes vs. paraphrase.

- W4. The faithfulness based activation editing strategy was introduced, but the results on it were not included in the paper.

**Questions:**

- Q1. Do you expect activation steering to be more or less effective for other models/architectures?
- Q2. Will you be releasing code/data for how faithfulness was calculated in this particular case?
- Q3. Do you expect your results to be consistent across how faithfulness metric was defined? So, for example, experimenting with faithfulness metric with paraphrasing vs. early answering strategy?

---

> ### Author Response · Authors · 2024-11-25
> **Response to Reviewer hqPm**
>
> Thank you for your thoughtful review of our paper. We are happy that you recognize our paper’s contributions towards connecting many disparate concepts and filling a significant gap in evaluating for interventions. We would like to address the weaknesses you pointed out and provide clarification on the questions raised.
>
> **W1/Q1: Do you expect activation steering to be more or less effective for other models/architectures?**
>
> We appreciate the suggestion to trial AE on other models. We are actively collecting results for Gemma and appreciate your patience.
>
> **W2: Finetuning with most faithful explanations vs finetuning by length?**
>
> Thank you for the suggestion to finetune according to length, which would provide good insights into how CoT length affects faithfulness across different domains. We kindly refer the reviewer to **Fig 11** (Appendix D.1, page 16), where we inspect test examples on which faithfulness remain the same/improved and observe that the average number of CoT reasoning steps used by each model generally increased (7/9 cases for ICL and 5/6 cases for FT). Models often invoked more granular CoT reasoning steps to improve faithfulness according to the early-answering metric. Mechanically, this is likely to increase the chances of a mismatch between intermediate and final answer probabilities (and thus the AOC). As such, we do not necessarily wish to optimize for faithfulness by learning shortcuts on the metric such as the above.
>
> Additionally, please see our **global response** for evidence of why finetuning is a fundamentally difficult challenge in the context of faithfulness, which we deem particularly relevant to this discussion.
>
> **Q2: Will you be releasing code/data for how faithfulness was calculated in this particular case?**
>
> Absolutely! We are happy to release our coding framework upon publication, which includes extensive pipelines that lay the groundwork for assessing faithfulness, using both OpenAI APIs and Hugging Face GPU implementations.
>
> **W3/Q3: Do you expect your results to be consistent across different faithfulness metrics?**
>
> This is a great question. We conducted additional experiments to evaluate faithfulness on the AQuA dataset using strategies proposed by Lanham et. al (2023). We looked at the *'Adding Mistakes'* and *'Paraphrasing'* strategies to measure faithfulness and find that both these measures are highly correlated with faithfulness from the *'Early Answering'* strategy, shown in **Fig 12** in the pdf (Appendix D.2, page 17). Our observation is also consistent with the findings reported by Lanham et. al. (2023).
>
> However, we do note that identifying fundamental challenges in applying interventions to one faithfulness measure (early answering) sheds much light on how optimizing for faithfulness can be severely limited. While extensive, unified analysis across further metrics would enhance the paper, we believe this belongs squarely in future work and does not detract significantly from our main illustrations regarding fundamental issues of faithfulness (namely, as per the global response, finetuning is a moving target, and ICL mimics word rather than learning intrinsic faithfulness, or learns other salient shortcuts such as increasing CoT granularity).
>
> **W4. The faithfulness based activation editing strategy was introduced, but the results on it were not included in the paper.**
>
> The results for activation editing strategy are presented in Section 4.2.3. Figure 10 shows how faithfulness and accuracy vary with the intervention hyper-parameters - number of heads $K$ and strength of intervention $\alpha$.
>
> Thank you once again for your insightful suggestions and comments. We are actively incorporating the above explanations to enhance the quality of our paper. We believe that we have addressed all the concerns, barring additional activation editing results. If there is any aspect that you feel has not been fully resolved, we would be happy to provide further information. If you are satisfied with our response, we would truly appreciate your consideration in raising your evaluation score.

---

> ### Author Response · Authors · 2024-11-27
> **Activation Editing results on Gemma-7B-IT and Mistral-7B-Instruct**
>
> Thank you for your patience.
>
> We appreciate the suggestion to trial Activation Editing strategy on other models. While we agree with the reviewer that effectiveness of activation steering can vary across models and attention mechanisms, Llama-3-8B-Instruct was a good starting point as it has proven to be effective in Li et al. (2023). We have performed activation editing (AE) experiments on Gemma-7B-IT and Mistral-7B-Instruct to make our results more robust as per the reviewer's requests. Please see page 22 and 23 of the updated manuscript. Gemma-7B-IT and Mistral-7B-Instruct are models of similar size as Llama-3-8B-Instruct but different architectures and attention mechanism. Gemma-7B-IT uses the well known self attention and Llama-3-8B uses grouped query attention (key and value heads are shared across a group of attention heads). In addition to grouped query attention, Mistral-7B-Instruct uses sliding window attention for efficient scaling to very long context windows.
>
> For all three datasets (AQuA, LogiQA, TruthfulQA), we ablate over the number of intervened heads ($K$) and the intervention strength $\alpha$. As was the case with the Llama model, none of the intervention configuration leads to improvement of both accuracy and faithfulness. There is a significant trade-off in accuracy as faithfulness improves, and intervening beyond 8 attention heads often leads to illegible responses.

---

> > ### Comment · Reviewer_hqPm · 2024-12-01
> >
> > Hi authors, thank you so much for such thoughtful discussions and additional results on activation editing on such a short notice. I also feel that my comments and questions were adequately addressed. I also think that the global responses were particularly helpful for expanding upon the faithfulness discussions on ICL.
> >
> > I believe that this paper addresses an important question of the consistency regarding faithfulness, and I personally think the authors considered many experiments from varying methods (finetuning, ICL, various metrics, various settings, across many tasks). However, my concern is mostly with the limited definition of the faithfulness itself (only one way of defining/measuring faithfulness). Overall, I maintain my favorable assessment of this paper.
> >
> > Thanks to the authors for an interesting paper, and please let me know if there are any misunderstanding on my part!

---

> > > ### Author Response · Authors · 2024-12-01
> > >
> > > Thank you for responding to our rebuttal! We are excited to hear that we were able to address your concerns. While we note the point of limited faithfulness definition by the reviewer, we would like to clarify that this is one of the key conclusions of our work, i.e., the faithfulness metrics derived from the current faithfulness definition in explainability literature do not result in improving the faithfulness of the reasoning generated by LLMs. Hence, we conclude that the community needs to propose better metrics and definitions of faithfulness.
> > >
> > > In summary, we would like to highlight again that the point raised by the reviewer is essentially the conclusion of our work. We will include this clarification in our paper's experiment and conclusion section. Please let us know if we could clarify any remaining concerns, and we would greatly appreciate it if you consider increasing your rating of our paper.

---

> ### Comment · Reviewer_hqPm · 2024-12-01
>
> I'd like to point out that, while the inconsistency in hardness is the conclusion of your work, only considering one metric of faithfulness is perhaps premature to arrive at that conclusion. Your rebuttal pointed this out as well,
> > However, we do note that identifying fundamental challenges in applying interventions to one faithfulness measure (early answering) sheds much light on how optimizing for faithfulness can be severely limited.
>
> I think it's less convincing if the definition of faithfulness is a moving target/not fully explored.

---

> > ### Author Response · Authors · 2024-12-01
> >
> > Thank you for the discussion, Reviewer hqPm. We appreciate you taking the time to discuss our rebuttal.

---

### Author Response · Authors · 2024-11-25
**Global Response**

Here, we demonstrate *why faithfulness is a fundamentally difficult property to optimize for* and *why existing intervention techniques provide limited success*.

## Why finetuning for model faithfulness is difficult

Finetuning involves gradient descent on some original model $\theta_0$ and loss, $L = \frac{1}{n}\sum_i \ell (x_i, y_i)$ to yield $\theta_1 = \arg\min_\theta L$. Most model properties operate on fixed ground truths (i.e., static $x_i, y_i$ pairs or completions).

**Finetuning for faithfulness shifts model internals**: Faithful explanations are by design defined w.r.t. a particular model. The faithfulness of an explanation is not fixed across models, unlike other properties such as accuracy, robustness. To demonstrate, we finetune Llama-3-8B-Instruct, $\theta_0$, on faithful demonstrations w.r.t. $\theta_0$, to yield $\theta_1$. We then use explanations from $\theta_1$ to measure faithfulness w.r.t. both $\theta_1$ and $\theta_0$.

| Dataset/Model | DU | DU (c) | DF | DF (c) | SU | SU (c) | SF | SF (c) |
|-|-|-|-|-|-|-|-|-|
| AQuA - Faith. wrt. original $\theta_0$ | **0.623** | **0.615** | **0.636** | **0.600** | **0.653** | **0.640** | **0.620** | **0.625** |
| AQuA - Faith. wrt. finetuned $\theta_1$ | 0.529 | 0.488 | 0.608 | 0.588 | 0.584 | 0.527 | 0.555 | 0.617 |
| |
| LogiQA - Faith. wrt. original $\theta_0$ | **0.405** | **0.383** | **0.409** | **0.406** | 0.427 | 0.435 | **0.443** | **0.411** |
| LogiQA - Faith. wrt. finetuned $\theta_1$ | 0.372 | 0.339 | 0.363 | 0.383 | **0.445** | **0.453** | 0.415 | 0.362 |
| |
| TruthfulQA - Faith. wrt. original $\theta_0$ | **0.239** | **0.222** | **0.252** | **0.224** | **0.263** | **0.264** | 0.253 | 0.247 |
| TruthfulQA - Faith. wrt. finetuned $\theta_1$ | 0.184 | 0.187 | 0.239 | 0.219 | 0.225 | 0.242 | **0.265** | **0.248** |

Our results show that faithfulness w.r.t. $\theta_1$ is notably lower than faithfulness w.r.t. $\theta_0$. This is intuitive since faithful demonstrations are provided from $\theta_0$. What this does strongly suggest is that (question, explanation) pairs that are faithful w.r.t. $\theta_0$, may no longer be guaranteed to be faithful w.r.t. a finetuned model $\theta_1$ (in our findings, they are decidedly more faithful w.r.t the original model). This ultimately supports our claim that **faithfulness is fundamentally difficult to target using finetuning.**

N.B. This is implied in Fig 13 of the Appendix, where finetuning shifts the model's early-answer probabilities despite yielding identical/semantically similar reasoning for each CoT step.

## Limitations of in-context learning (ICL)

1. **ICL mimics explicit words in its examples, instead of learning implicit faithfulness properties:** We measure Jensen-Shannon Divergence between the in-context examples and the CoT response from a) zero-shot (ZS) and b) ICL. Lower is more similar (bolded). In almost all cases, word distributions from ICL CoT are more similar to the in-context examples than ZS CoT words.

Dataset | DU | DU (c) | DF | DF (c) | SU | SU (c) | SF | SF (c) |
|-|-|-|-|-|-|-|-|-|
AQuA (Examples vs ZS) | **0.4658** | **0.4589** | **0.4639** | **0.4779** | **0.4519** | **0.4517** | **0.4620** | **0.4769** |
AQuA (Examples vs ICL) | 0.4664 | 0.4664 | 0.4690 | 0.4873 | 0.4617 | 0.4602 | 0.4675 | 0.4870 |
| |
LogiQA (Examples vs ZS) | 0.5339 | **0.5224** | **0.5326** | **0.5576** | **0.5282** | **0.5302** | **0.5263** | **0.5447** |
LogiQA (Examples vs ICL) | **0.5288** | 0.5288 | 0.5386 | 0.5665 | 0.5364 | 0.5353 | 0.5343 | 0.5555 |
| |
TruthfulQA (Examples vs ZS) | **0.5162** | **0.5138** | **0.5199** | **0.5190** | **0.5095** | **0.5185** | **0.5116** | **0.5119** |
TruthfulQA (Examples vs ICL) | 0.5208 | 0.5208 | 0.5259 | 0.5239 | 0.5221 | 0.5228 | 0.5237 | 0.5180 |

2. **Optimizing for faithfulness can have tradeoffs with accuracy.** While ICL improved faithfulness in some approaches, there is often a trade off in accuracy as shown in Figs 5, 7, 15. This is due to the metric incentivizing changes in label predictions deep into reasoning (Fig 13). Observe the original model's CoT on the number of human finger bones that leads to the correct answer. In this case, ICL has induced reasoning that sways the model's idea of the final answer throughout. In particular, our observations tend to reveal that a late change in reasoning from the model is a typical aspect of faithful CoT that can ultimately get optimized. Fig 3 demonstrates how faithfulness can be at odds with accuracy as a result of this.

### Remarks

We thank the reviewers for requesting more information on why faithfulness is fundamentally difficult to achieve. We provide further results in the tables above, from carefully designed experiments, to support the case that existing techniques require rethinking specifically for the case of faithfulness. We appreciate your patience as we actively incorporate the above results and transfer the appendix analyses to the main text.

---

### Meta-Review · Area_Chair_F1oD · 2024-12-21

**Metareview:**

The paper examined the Chain-of-Thought (CoT) behavior in large language models (LLMs) through experiments involving in-context learning, fine-tuning, and activation editing. This work combines the three aforementioned techniques to improve LLMs in producing what is referred to as faithful CoT reasoning and rational explanations to the delivered output. The authors argue that the training process of these models does not prioritize faithfulness, which contributes to the generation of more self-aware content. The paper targets an important and timely challenge in LLM with detailed experiment design. While there are several major concerns remain regarding the novelty of the proposed method and the investigation of the empirical results. The proposed solution is combining applied techniques in LLMs, which render the contribution incremental and straightforward. Technically, the contribution lacks in rigor, and many of the applied strategies are not formally justified. Furthermore, there is lack of insights on the empirical results. For instance, why faithfulness is difficult to learn, either in the form of mathematical theorems, or carefully designed experiments would be helpful. And it remains unclear why “early answering” is a reasonable metric for measuring faithfulness, particularly for black-box LLMs. Given the above reasons, after discussion with the reviewers, we would encourage the authors to expand their analysis and propose more innovative approaches in future iterations of this work for resubmission.

**Additional Comments On Reviewer Discussion:**

Though part of reviewers' concerns have been resolved during rebuttal, the main concerns from reviewers remain regarding the novelty of the proposed method and the investigation of the empirical results. The proposed solution is combining applied techniques in LLMs, which render the contribution incremental and straightforward. Furthermore, there is lack of insights on the empirical results. Given the above reasons, after discussion with the reviewers, we would encourage the authors to expand their analysis and propose more innovative approaches in future iterations of this work for resubmission.

---

### Decision · Program_Chairs · 2025-01-22

Reject